# YOD1/TRAF6 association balances p62-dependent IL-1 signaling to NF-κB

Gisela Schimmack[1], Kenji Schorpp[2], Kerstin Kutzner[1], Torben Gehring[1], Jara Kerstin Brenke[2], Kamyar Hadian[2], Daniel Krappmann[1]*

[1]Research Unit Cellular Signal Integration, Institute of Molecular Toxicology and Pharmacology, Helmholtz Zentrum München - German Research Center for Environmental Health, Neuherberg, Germany; [2]Assay Development and Screening Platform, Institute of Molecular Toxicology and Pharmacology, Helmholtz Zentrum München - German Research Center for Environmental Health, Neuherberg, Germany

**Abstract** The ubiquitin ligase TRAF6 is a key regulator of canonical IκB kinase (IKK)/NF-κB signaling in response to interleukin-1 (IL-1) stimulation. Here, we identified the deubiquitinating enzyme YOD1 (OTUD2) as a novel interactor of TRAF6 in human cells. YOD1 binds to the C-terminal TRAF homology domain of TRAF6 that also serves as the interaction surface for the adaptor p62/Sequestosome-1, which is required for IL-1 signaling to NF-κB. We show that YOD1 competes with p62 for TRAF6 association and abolishes the sequestration of TRAF6 to cytosolic p62 aggregates by a non-catalytic mechanism. YOD1 associates with TRAF6 in unstimulated cells but is released upon IL-1$\beta$ stimulation, thereby facilitating TRAF6 auto-ubiquitination as well as NEMO/IKKγ substrate ubiquitination. Further, IL-1 triggered IKK/NF-κB signaling and induction of target genes is decreased by YOD1 overexpression and augmented after YOD1 depletion. Hence, our data define that YOD1 antagonizes TRAF6/p62-dependent IL-1 signaling to NF-κB.

**\*For correspondence:** daniel.krappmann@helmholtz-muenchen.de

**Competing interests:** The authors declare that no competing interests exist.

## Introduction

The inflammatory cytokine interleukin-1 (IL-1) activates canonical IκB kinase (IKK)/NF-κB signaling upon binding to the IL-1 receptor (IL-1R). IL-1 induces recruitment of MYD88 and IRAK proteins to the IL-1R to form the Myddosome (*Cohen, 2014*). In turn, IRAK1, 2 and 3 interact with the E3 ligase TNF-receptor associated factor 6 (TRAF6), which is an essential component to initiate IL-1R downstream signaling (*Ye et al., 2002*; *Cao et al., 1996*). TRAF6 bridges the Myddosome to TAK1 that acts as an IKK$\beta$ upstream kinase on the route to NF-κB (*Lomaga et al., 1999*; *Wang et al., 2001*; *Sato et al., 2005*).

TRAF6 belongs to the TRAF family of proteins that share a C-terminal TRAF region, consisting of coiled-coil and MATH (Meprin and TRAF homology) domains, which are needed for oligomerization and adaptor function (*Ha et al., 2009*). The N-terminal RING and Zinc Finger1 (Z1) domain confers ubiquitin ligase activity. In response to IL-1 stimulation, TRAF6 in conjunction with the E2 enzyme UBC13/UEV1A catalyzes the attachment of lysine (K)63-linked ubiquitin chains to substrate proteins, including TRAF6 itself, IRAK1, TAK1 and the non-catalytic IKK complex component NEMO/IKKγ (*Deng et al., 2000*; *Conze et al., 2008*; *Fan et al., 2010*; *Yamazaki et al., 2009*). Reconstitution experiments reveal that TRAF6 E3 ligase activity is critical for activation of NF-κB signaling in response to IL-1 (*Lamothe et al., 2007*; *Walsh et al., 2008*). Since TRAF6 overexpression alone is sufficient to strongly activate NF-κB (*Cao et al., 1996*), its activity needs to be tightly controlled by positive and negative regulators.

The atypical PKC-interacting protein p62 (also called Sequestosome-1; SQSTM-1) is essential for TRAF6-dependent canonical NF-κB signaling and activation in response to IL-1 stimulation (*Sanz et al., 2000*). p62 has also been implicated in other TRAF6-dependent signaling pathways emanating from CD40, RANK or NGF (*Wooten et al., 2005*; *Durán et al., 2004*; *Seibold and Ehrenschwender, 2015*). TRAF6 is recruited to p62 aggresomes and p62 promotes TRAF6 E3 ligase activity to enhance auto- and substrate ubiquitination (*Sanz et al., 2000*; *Zotti et al., 2014*; *Wooten et al., 2005*). However, p62 also recruits the deubiquitinating enzyme (DUB) CYLD (Cylindromatosis) that acts as negative regulator of NF-κB signaling (*Jin et al., 2008*; *Wooten et al., 2008*). CYLD cleaves K63-linked ubiquitin chains conjugated to TRAF6 and its substrates (*Yoshida et al., 2005*; *Reiley et al., 2007*), revealing that it counteracts signaling by a catalytic mechanism. Besides CYLD, the ubiquitin editing enzyme A20 counterbalances TRAF6 activity. After pro-longed IL-1 stimulation A20 binds to TRAF6 to prevent TRAF6/UBC13 interaction by a process independent of its DUB activity (*Shembade et al., 2010*).

Here, we report on the identification of the OTU (ovarian tumor) DUB family member YOD1 (homolog of yeast OTU1; OTUD2, DUBA8) as a new interactor of TRAF6. Originally, the yeast homolog OTU1 was shown to act as a cofactor of the hexameric AAA-ATPase Cdc48/p97 for protein processing (*Rumpf and Jentsch, 2006*). In mammalian cells, YOD1 facilitates protein quality control by Valosin-containing protein (VCP)/p97 at the endoplasmic reticulum (ER) through the ER-associated protein degradation (ERAD) pathway (*Claessen et al., 2010*; *Ernst et al., 2009*). We find that YOD1 directly associates with TRAF6 and competes with p62 for TRAF6 binding and activation. YOD1 is released from TRAF6 upon IL-1 stimulation and YOD1 depletion enhances canonical NF-κB activation. These results define YOD1 as novel negative regulator of TRAF6/p62-triggered IL-1 signaling.

## Results

### YOD1 associates with the C-terminal MATH domain of TRAF6

To identify regulators of TRAF6, we searched for interaction partners by yeast-two-hybrid (Y2H). We discovered a novel interaction of YOD1 with full length TRAF6, but not with a fragment comprising the catalytic RING-Zinc Finger1 (Z1) domain (*Figure 1A*; *Figure 1—figure supplement 1A*). Originally, mammalian YOD1 and its yeast homolog OTU1 were identified as co-factor of the AAA-ATPase p97/Cdc48 (*Rumpf and Jentsch, 2006*; *Ernst et al., 2009*), which was also confirmed in the Y2H (*Figure 1A*). Besides the binding to YOD1, TRAF6 self-associated and bound to the E2 enzyme UBC13 and the ubiquitin editing enzyme A20, but not to p97 (*Figure 1—figure supplement 1B*). To obtain data on the selectivity of YOD1/TRAF6 interaction, we checked binding of YOD1 to a panel of E3 ligases as well as other proteins involved in regulatory ubiquitination by Y2H (*Figure 1—figure supplement 1C*). Despite some weak interaction with cIAP2 and SHARPIN, within this panel YOD1 was quite selectively binding to p97 and TRAF6. Protein expression was determined by Western Blotting and functionality of some proteins in yeast was confirmed by verifying known interactions of selected constructs (*Figure 1—figure supplement 1D and E*). Of note, even though expression of some E3 ligases was barely detectable, the interaction with known partners (e.g. HOIP with OTULIN or cIAP1/2 with UBC13) was readily detectable in the growth assay. To confirm that TRAF6 and YOD1 are directly associating and to narrow down the YOD1 binding site on TRAF6, we expressed and purified recombinant YOD1 and TRAF6 proteins and performed pull-down (PD) experiments (*Figure 1B* and *Figure 1—figure supplement 2A*). Clearly, YOD1 was binding to the C-terminal MATH (346 -504) but not to the N-terminal RING-Z1 (50–159) domain of TRAF6. In addition, we confirmed binding of recombinant YOD1 to p97 by GST-PD (*Figure 1—figure supplement 2B*). In a triple PD experiment we neither found that YOD1-p97 association was prevented by TRAF6 nor that p97 did influence interaction of YOD1 to the TRAF6 MATH domain, revealing that both proteins can bind to YOD1 independently (*Figure 1—figure supplement 2C*).

To investigate whether YOD1 and TRAF6 also interact in cells, we co-expressed full length FLAG-YOD1 and HA-TRAF6 in HEK293 cells. Co-immunoprecipitations (IP) using anti-HA or anti-FLAG antibodies confirmed the interaction of TRAF6 and YOD1 (*Figure 1C–E*). Congruent with the Y2H and PD results, YOD1 bound to the C-terminal MATH domain, but not to the N-terminal RING-Z1-Z4 region of TRAF6 (*Figure 1D*). On the side of YOD1, the N-terminal UBX domain was essential to mediate TRAF6 interaction (*Figure 1E*, scheme *Figure 1F*). Sequence comparison indicated the

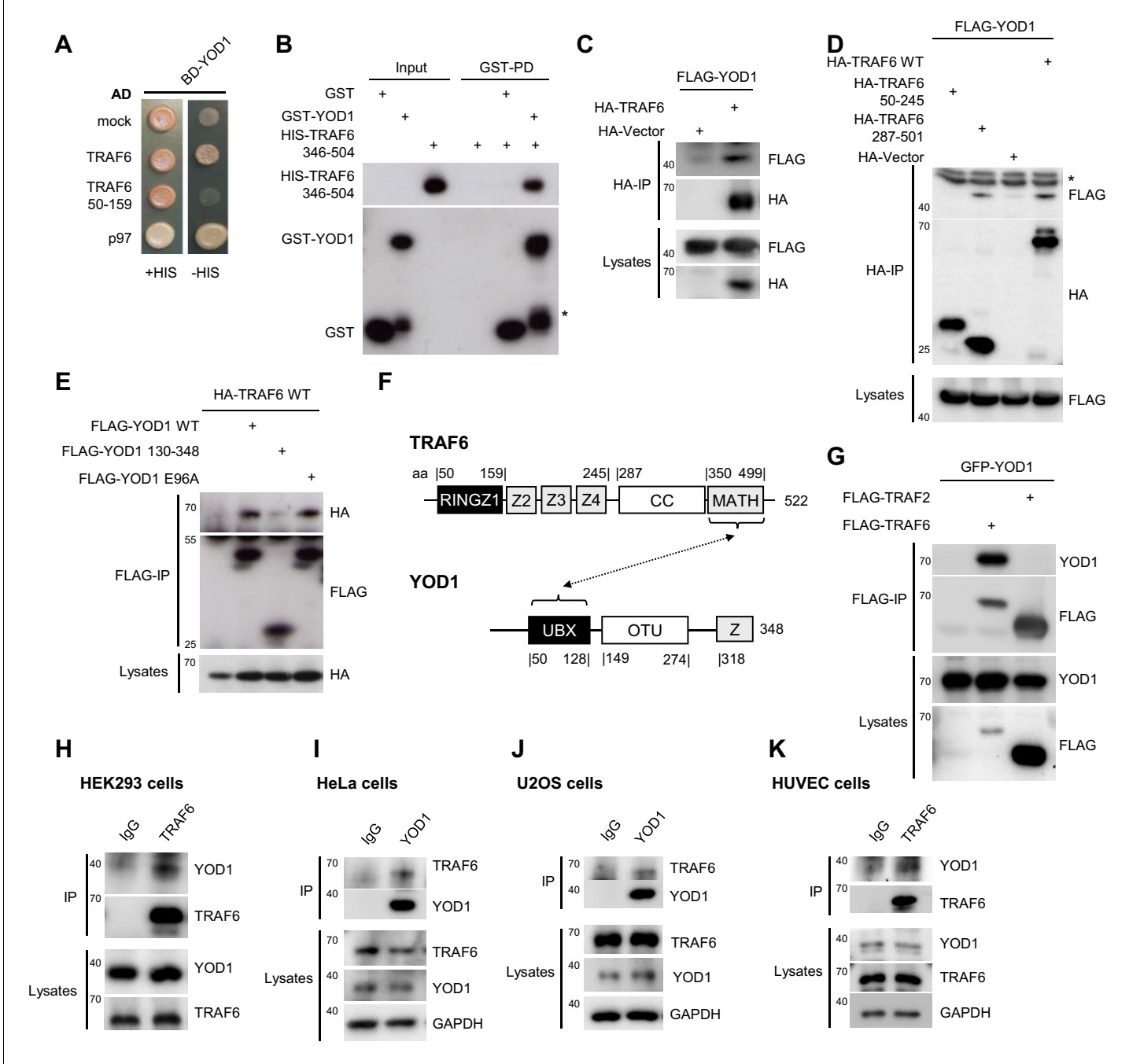

**Figure 1.** YOD1 interacts with the C-terminal MATH domain of TRAF6. (**A**) YOD1 interacts with full length TRAF6 and p97 in a yeast two hybrid assay. Activating domain (AD) and binding domain (BD) fusion constructs were co-transformed as indicated and growth was monitored on -LEU-TRP control (+HIS) and -HIS-LEU-TRP (−HIS) plates. (**B**) The MATH domain of TRAF6 is sufficient for interaction with YOD1 in vitro. GST-PD were performed with recombinant GST-YOD1 or GST and C-terminal HIS-TRAF6 MATH (346-504) and analyzed by Western Blotting. Asterisk indicates GST-YOD1 truncation product. (**C**) YOD1 and TRAF6 interact in cells. HEK293 cells were co-transfected with FLAG-YOD1 and HA-TRAF6 or HA-control vector and co-IP was carried out using anti-HA antibodies and analyzed by Western Blot. Asterisk depicts IgGs. (**D**) YOD1 binds to the C-terminus of TRAF6. YOD1 was co-expressed with TRAF6 deletion or control constructs as indicated. Experiment was performed as in (**C**). Asterisk depicts IgGs. (**E**) TRAF6 binds to the UBX domain of YOD1. HA-TRAF6 was co-expressed with FLAG-YOD1, FLAG-YOD1 ΔUBX (130-348) or FLAG-YOD1 E96A. Experiment was performed using anti-FLAG IP as in (**C**). (**F**) Schematic summary of the domains required for YOD1/TRAF6 interaction as determined by co-IPs and PDs (compare also *Figure 3C*). (**G**) YOD1 does not bind to TRAF2. After transfection of GFP-YOD1 and Flag-TRAF2 or Flag-TRAF6 the experiment was performed using anti-FLAG IP as in **C**. (**H − K**) Endogenous interaction of YOD1 and TRAF6. HEK293 (**H**), HeLa (**I**), U2OS (**J**) or HUVEC (**K**) cells were subjected to TRAF6 (**H and K**) or YOD1 (**I and J**) IP as indicated. IgG IP was used as control. Co-precipitation of YOD1 or TRAF6 was analyzed by Western Blotting.

*Figure 1 continued on next page*

*Figure 1 continued*

The following figure supplements are available for figure 1:

**Figure supplement 1.** TRAF6/YOD1 interaction in yeast.

**Figure supplement 2.** TRAF6/YOD1 interaction is not influenced by p97.

**Figure supplement 3.** Analysis of YOD1/TRAF6 binding in cells.

existence of a putative TRAF6 interaction motif (TIM) for MATH interactors within the UBX domain (PXEXXAr/Ac) (*Ye et al., 2002*) (*Figure 1—figure supplement 3A*). However, neither exchange of the conserved glutamic acid to alanine (YOD1 E96A) nor even more profound mutations of the putative TRAF6 binding motif abolished YOD1 association (*Figure 1E* and *Figure 1—figure supplement 3B*), indicating that binding of TRAF6 MATH to YOD1 UBX domain is not mediated through a typical TIM. To assess the selectivity of YOD1/TRAF6 interaction, we compared association of YOD1 to TRAF2 and TRAF6 in HEK293 cells (*Figure 1G*). We did not detect YOD1-TRAF2 binding, indicating a selectivity of YOD1 for association with TRAF6.

Next, we performed IPs of endogenous TRAF6 and YOD1 in HEK293, HeLa and U2OS cell lines as well as in primary human umbilical vein endothelial cells (HUVEC) (*Figure 1H–K*). Indeed, in all cell lines and primary HUVEC YOD1 was specifically co-precipitating with endogenous TRAF6, as validated by either TRAF6 or YOD1 IP. Further, specificity of TRAF6/YOD1 interaction was confirmed by the decreased co-precipitation of YOD1 in TRAF6 knock-down HeLa cells (*Figure 1—figure supplement 3C*). We compared the expression levels of TRAF6 and YOD1 in different cell lines (HeLa, HEK293, U2OS and PC3 cells) and tested, if there was a correlation between expression and association (*Figure 1—figure supplement 3D*). TRAF6/YOD1 binding was visible in all cells and there was a tendency that more TRAF6/YOD1 association was observed in cells that expressed more TRAF6 (U2OS and HEK293). Since YOD1 has been described as a cellular co-factor of p97, we checked for YOD1/TRAF6 interaction in p97 knock-down HEK293 cells and found that binding of YOD1 to TRAF6 was not significantly altered upon p97 depletion (*Figure 1—figure supplement 3E*). Thus, cellular and in vitro binding studies identify the deubiquitinating enzyme YOD1 as a direct interaction partner of the E3 ligase TRAF6.

## YOD1 co-localizes with TRAF6 in cytosolic speckles and competes with p62 for TRAF6 association

To gain insights into the role of YOD1/TRAF6 interaction, we determined the cellular localization of both proteins upon overexpression in U2OS and HeLa cells by confocal fluorescence microscopy. Whereas RFP-TRAF6 was distributed in small dots in the cytoplasm but not in the nucleus, GFP-YOD1 and catalytically inactive GFP-YOD1 C160S were evenly dispersed in the cytoplasm and nucleus with some accumulations in or around the nucleus (*Figure 2A* and *Figure 2—figure supplement 1A*). Upon co-expression, GFP-YOD1 and RFP-TRAF6 were to a large extent co-localizing to cytosolic speckles, indicating that the proteins interact inside the cell and can form larger clusters (*Figure 2B* and *Figure 2—figure supplement 1B and C*). To confirm co-localization, we plotted fluorescence intensities (FI) of RFP-TRAF6 and GFP-YOD1 through spot containing sections and show that the peaks of highest RFP and GFP FI overlap (*Figure 2B* and *Figure 2—figure supplement 1C*). For a quantitative analysis of co-localization we performed automated image analysis of ~200 cells and determined the FI of RFP-TRAF6 and GFP-YOD1 in the GFP and RFP clusters, respectively (*Figure 2—figure supplement 1D*). Compared to the background, the RFP-TRAF6 signal was enriched in GFP-YOD1 spots and vice versa the GFP-YOD1 signal was enhanced in RFP-TRAF6 spots, clearly suggesting co-localization of TRAF6 and YOD1 in the clusters. Interaction of TRAF6 and YOD1 did not rely on YOD1 catalytic activity, because like YOD1 WT, the DUB mutant YOD1 C160S was co-localizing (*Figure 2B* and *Figure 2—figure supplement 1C*) and co-precipitating with TRAF6 (*Figure 2C*).

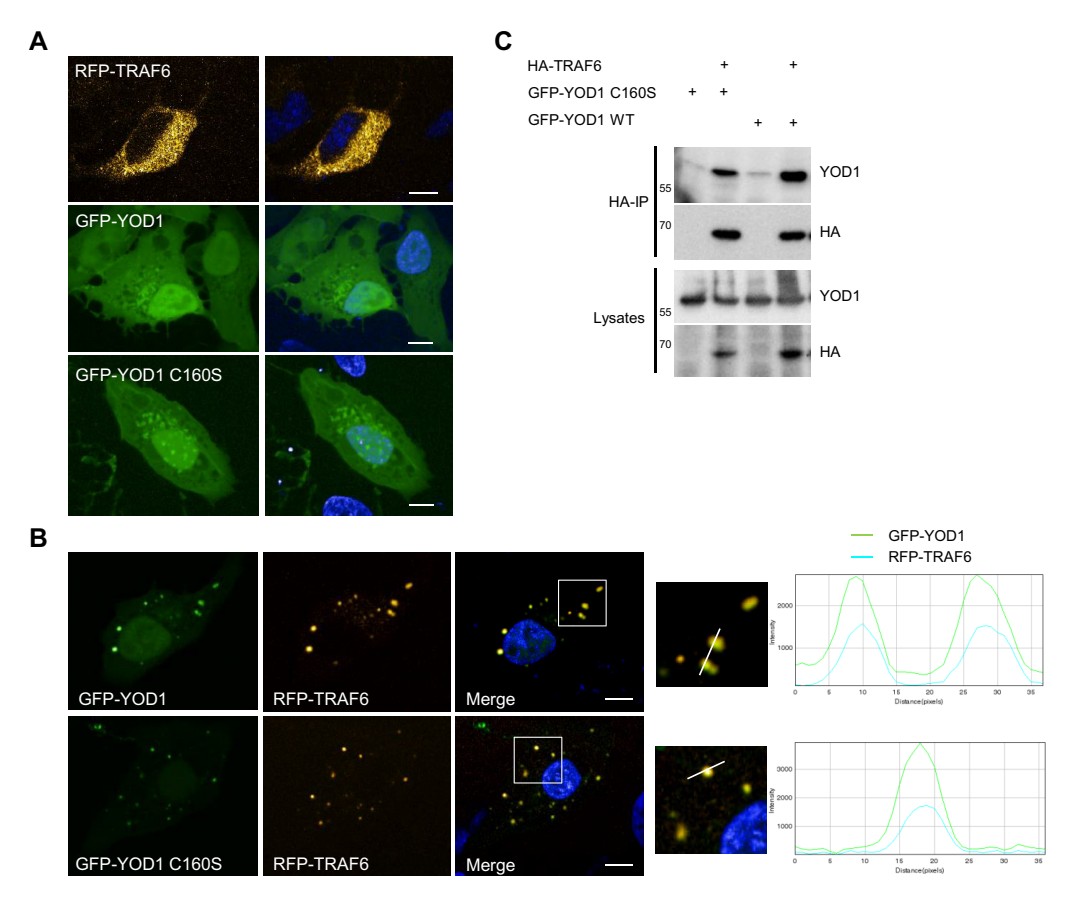

**Figure 2.** YOD1 co-localizes with TRAF6 in cytosolic speckles. (**A**) Diffuse localization of TRAF6 and YOD1 upon individual expression. RFP-TRAF6, GFP-YOD1 or GFP-YOD1 C160S were overexpressed in U2OS cells and localization was analyzed by confocal fluorescence microscopy. (**B**) YOD1 and TRAF6 co-localize in cytosolic speckles upon co-expression. The co-localization is independent of YOD1 catalytic activity. GFP-YOD1 (WT or C160S) and RFP-TRAF6 were co-transfected in U2OS cells and localization was analyzed as in (**A**). Enlargement of boxed area is shown next to Merge. Plot Profile analysis was conducted to quantify fluorescence intensities and to monitor co-localization along the white line. (**C**) TRAF6 interacts with YOD1 independent of its catalytic activity. HEK293 cells were co-transfected with HA-TRAF6, GFP-YOD1 WT and GFP-YOD1 C160S constructs as indicated. Co-IP was carried out using anti-HA antibodies and analyzed by Western Blot. Merged pictures include nuclear staining with Hoechst 33342. Scale bars depict 10 μM (**A** and **B**).

The following figure supplement is available for figure 2:

**Figure supplement 1.** YOD1/TRAF6 co-localization.

The staining and localization of the YOD1/TRAF6 speckles displayed similarities to cytoplasmic aggregates termed sequestosomes that have been described for the TRAF6 interactor p62/Sequestosome-1 (*Sanz et al., 2000*; *Seibenhener et al., 2004*; *Wang et al., 2006*). As expected, we observed similar aggregates when Crimson-p62 was expressed in U2OS cells (*Figure 3A*) and TRAF6 was recruited to the punctuated p62 sequestosomes in U2OS cells as evident from plotted FI of CFP-p62 and RFP-TRAF6 spots as well as co-clustering of FI as measured by automated image analysis (*Figure 3B* and *Figure 3—figure supplement 1A*). In contrast, GFP-YOD1 was not co-localizing with Crimson-p62 aggregates. However, it appeared that p62 staining was slightly more diffuse in YOD1 expressing cells, hinting at an indirect effect of YOD1 on the formation of p62 sequestosomes (*Figure 3B* and *Figure 3—figure supplement 1B*). We confirmed by co-IP that p62 binds to FLAG-TRAF6, but not to FLAG-YOD1 in HEK293 cells (*Figure 3—figure supplement 1C*).

The MATH domain of TRAF6 serves as interaction and oligomerization platform and thus it is critically involved in regulating TRAF6 functions in NF-κB signaling (*Ye et al., 2002*; *Walsh et al., 2015*).

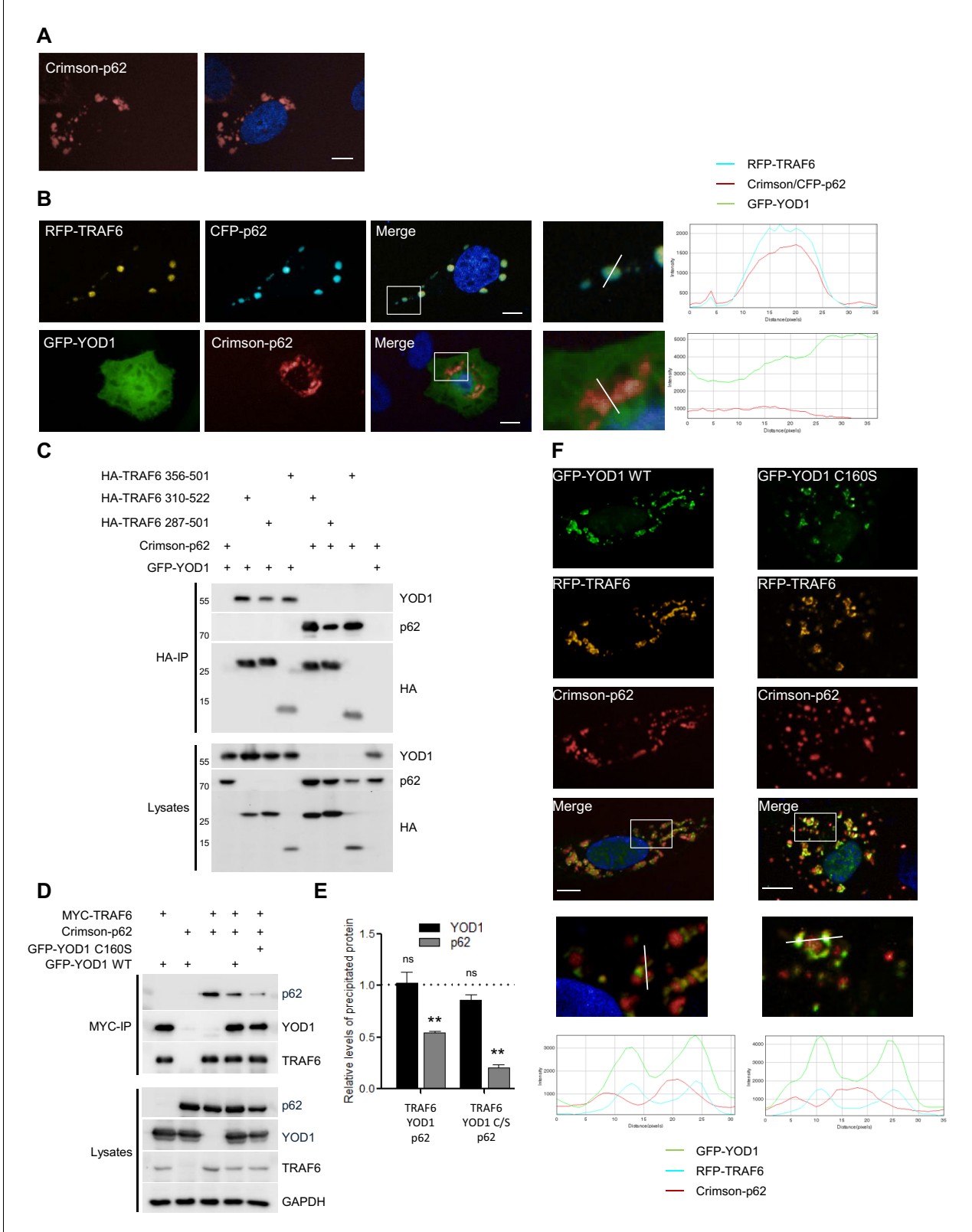

**Figure 3.** YOD1 competes with p62 for binding to TRAF6 and recruitment to sequestosomes. (**A**) p62 localizes to sequestosomes. Crimson-p62 was transfected in U2OS cells and localization was analyzed by confocal fluorescence microscopy. (**B**) TRAF6, but not YOD1, is recruited to p62-containing aggregates. RFP-TRAF6 and CFP-p62 or GFP-YOD1 and Crimson-p62 were co-transfected in U2OS cells and localization was analyzed as in (**A**). Enlargement of boxed area is shown next to Merge. Plot Profile analysis was conducted to quantify fluorescence intensities and monitor co-localization

*Figure 3 continued on next page*

*Figure 3 continued*

along the white line. (**C**) YOD1 and p62 bind to the C-terminal MATH domain of TRAF6. HEK293 cells were co-transfected with the indicated constructs. Co-IP was carried out using anti-HA antibodies and analyzed by Western Blot. (**D** and **E**) YOD1 impedes p62/TRAF6 interaction. (**D**) HEK293 cells were co-transfected with GFP-YOD1 WT, GFP-YOD1 C160S, Crimson-p62 and MYC-TRAF6 constructs as indicated. co-IP was carried out using anti-MYC antibodies and analyzed by Western Blot. (**E**) For quantification of *Figure 3D* and two additional experiments, amounts of YOD1 or p62 bound to TRAF6 in double transfected cells were set to 1. Changes in binding upon co-expression of all three proteins were measured using LabImage 1D software. Data depict the mean and standard error of the mean (SEM) of three independent experiments. Significance for the decrease p62 and YOD1 versus control was evaluated using Student's t-test (\*\*p<0,01; ns = not significant). (**F**) YOD1 WT and C160S diminish recruitment of TRAF6 to p62 aggregates. GFP-YOD1 WT or C160S, respectively, RFP-TRAF6 and Crimson-p62 were co-expressed in U2OS cells and localization was analyzed as in (**A**). Enlargement of boxed area is shown below Merge. Plot Profile analysis was conducted along the white line. Merged pictures include nuclear staining with Hoechst 33342. Scale bars depict 10 μM (**A**, **B** and **F**).

The following figure supplements are available for figure 3:

**Figure supplement 1.** TRAF6, but not YOD1, is interacting with p62.
**Figure supplement 2.** YOD1/p62 competition for TRAF6 binding.

By directly comparing the binding of YOD1 and p62 to TRAF6 in HEK293 cells we could confirm that both proteins are associating with the C-terminal MATH domain (*Figure 3C*). Therefore, we investigated whether YOD1 and p62 are binding simultaneously to TRAF6 and potentially forming a tripartite complex or if binding of the two MATH domain interactors is mutually exclusive and possibly even competitive. Indeed, co-IP experiments using MYC-TRAF6 together with Crimson-p62 and GFP-YOD1 revealed that YOD1 was able to inhibit the association of TRAF6 and p62, while p62 did not alter the binding of YOD1 to TRAF6 (*Figure 3D and E*). Inhibition of p62/TRAF6 binding was independent of YOD1 catalytic activity, suggesting that YOD1 and p62 are competing for an association with TRAF6 independent of YOD1 DUB activity. Since p62/TRAF6 as well as YOD1/TRAF6 form cytosolic aggregates, we carefully analyzed their localization by confocal microscopy when all three proteins (RFP-TRAF6, Crimson-p62 and GFP-YOD1) were co-expressed in U2OS and HeLa cells (*Figure 3F* and *Figure 3—figure supplement 2A and B*). Even though all three proteins were found in cytoplasmic speckles, the merged images especially at a higher magnification indicated that while YOD1 and TRAF6 were co-localizing within clusters as seen in the absence of Crimson-p62 (see *Figure 2B*), p62 in contrast was in the vicinity, but largely excluded from YOD1/TRAF6 aggregates. The same distribution was seen when catalytically inactive YOD1 C160S was expressed in U2OS cells (*Figure 3F* and *Figure 3—figure supplement 2B*). In line, the plotted FI showed co-localization of RFP-TRAF6 and GFP-YOD1 but not Crimson-p62 peaks in all analyses. Also, automated image analysis of FI in a larger number of cells indicates a stronger enrichment of RFP-TRAF6 signal in GFP-YOD1 spots compared to Crimson-p62 spots and no co-localization beyond background of Crimson-p62 in RFP-TRAF6 or GFP-YOD1 clusters (*Figure 3—figure supplement 2C*). Thus, co-IP and co-localization studies suggest that YOD1, p62 and TRAF6 are not found in a trimeric complex, but that YOD1 and p62 compete for TRAF6 association and that YOD1 can interfere with the recruitment of TRAF6 to p62 sequestosomes by a non-catalytic mechanism.

## YOD1 antagonizes IL-1β induced IKK/NF-κB signaling

Since TRAF6 and p62 are acting in concert to promote IL-1R induced NF-κB activation (*Sanz et al., 2000*; *Zotti et al., 2014*), we wanted to determine the influence of YOD1 on NF-κB activation after IL-1β stimulation. By lentiviral transduction we generated HeLa cells that inducibly overexpress YOD1 WT or YOD1 C160S (*Figure 4A*). We used a doxycycline (DOX)-inducible expression system and generated a HeLa cell population that stably expresses the transcriptional repressor tTR-KRAB together with dsRed (*Wiznerowicz and Trono, 2003*) (*Figure 4—figure supplement 1A*). Under DOX/tTR-KRAB control, we then co-expressed YOD1 and GFP using the co-translational processing site T2A. Sorting by FACS yielded homogenous populations of GFP expressing cells after DOX treatment (*Figure 4—figure supplement 1B*), correlating with YOD1 overexpression in the infected cells (*Figure 4B*). We consistently found that expression of catalytically inactive YOD1 C160S was substantially lower than YOD1 WT, potentially indicating toxic effects of high overexpression of the mutant

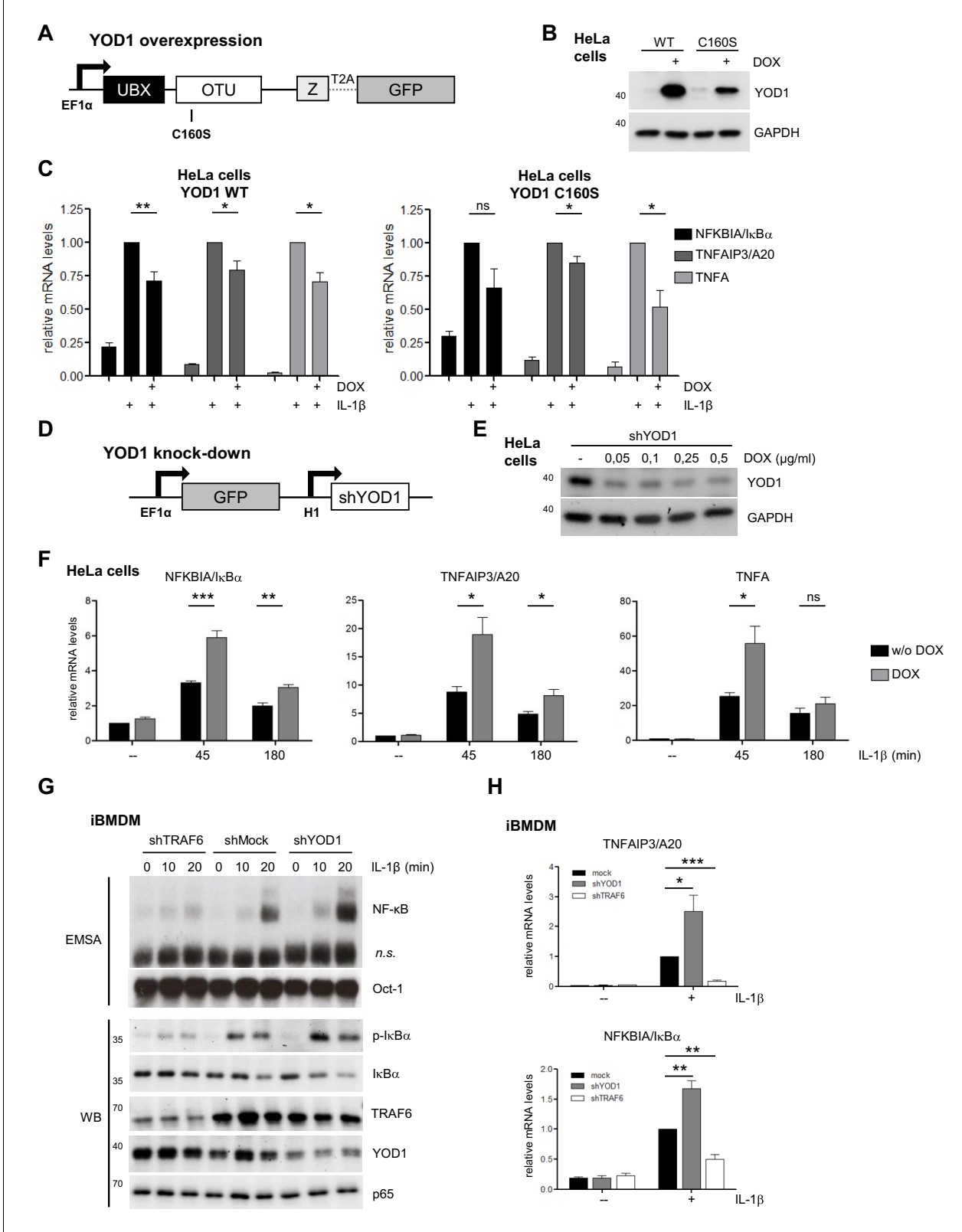

**Figure 4.** YOD1 is a negative regulator of IL-1β-induced NF-κB signaling. (**A**) Schematic representation of YOD1 overexpression constructs. YOD1 WT or C160S and GFP were co-expressed using T2A site under the control of EF1α promoter, which in turn is DOX/tTR-KRAB-controlled. (**B**) YOD1 WT and YOD1 C160S are overexpressed upon doxycycline (DOX) treatment of lentivirally transduced HeLa cells. Transduced cells were grown in DOX containing medium for 72 hr and after cell lysis subjected to Western Blotting. (**C**) YOD1 WT (left panel) or C160S (right panel) overexpression

*Figure 4 continued on next page*

*Figure 4 continued*

diminishes NF-κB target gene expression. Infected HeLa cells were treated with DOX for 72 hr and stimulated with IL-1β for 60 min. Expression of indicated transcripts was analyzed by qRT-PCR. Bars show mean and standard error of the mean (SEM) of five independent experiments. (D) Schematic representation of YOD1 shRNA construct. GFP and shYOD1 were expressed under control of EF1α and H1 promoter, respectively. Both promoters are DOX/tTR-KRAB-controlled. (E) YOD1 protein levels are reduced in shYOD1 cells. Cells were treated for 72 hr with 0,05–0,5 µg/ml DOX as indicated and YOD1 knock-down was analyzed by Western Blot. (F) YOD1 knock-down results in enhanced NF-κB target gene expression. shYOD1-infected HeLa cells were treated with DOX for 72 hr and stimulated with IL-1β for the indicated time points. RNA was isolated and transcripts were analyzed by qRT-PCR as indicated. Bars show mean and SEM of four independent experiments. (G) TRAF6 and YOD1 exert opposing effects on NF-κB signaling and activation in iBMDM. iBMDM transduced with control shMock, shTRAF6 or shYOD1 were stimulated with IL-1β as indicated. NF-κB and Oct-1 (control) DNA binding was assessed by EMSA (n.s. = non-specific band). IκBα phosphorylation, degradation and knock-down efficiencies were analyzed by Western Blotting. (H) YOD1 knock-down promotes, while TRAF6 depletion impairs NF-κB target gene expression in iBMDM. iBMDM transduced as in (G) were stimulated with IL-1β for 45 min. Transcript levels were analyzed by qRT-PCR as indicated. Bars show mean and SEM of seven independent experiments. Significance was evaluated using Student's t-test (*p<0,05; **p<0,01; ***p<0001; ns = not significant).

The following figure supplement is available for figure 4:

**Figure supplement 1.** Lentiviral transduction and DOX control treatment of HeLa cells.

(*Figure 4B*). To address if overexpression of YOD1 impacts on NF-κB activation, we measured by quantitative (q)RT-PCR the expression of the well-defined NF-κB target genes *NFKBIA/IκBα* , *TNFAIP3/A20* and *TNFA* in response to IL-1β in the absence or presence of overexpressed YOD1 (minus or plus DOX, respectively) (*Figure 4C*). While DOX treatment alone did not significantly alter expression of these genes in HeLa parental cells (*Figure 4—figure supplement 1C*), expression of YOD1 WT or C160S caused a significant decline in NF-κB target gene induction after IL-1β stimulation, indicating that YOD1 can antagonize IL-1R triggered NF-κB signaling independent of its catalytic activity.

To validate our finding about a negative regulatory role of YOD1 for IL-1R signaling to NF-κB, we knocked-down endogenous YOD1. Again, we used a lentiviral transduction system to generate cells that stably integrate the YOD1 shRNA and GFP marker gene, whose expression is under control of tTR-KRAB/DOX (*Figure 4D*). After lentiviral transduction of HeLa cells, DOX treatment led to strong and homogenous GFP expression, which correlated with a decrease in YOD1 protein expression upon increasing DOX concentrations (*Figure 4E – Figure 4—figure supplement 1D*). Again, we analyzed expression of NF-κB target genes upon IL-1β stimulation in YOD1 expressing (minus DOX) or depleted (plus DOX) HeLa cells (*Figure 4F*). In line with a negative regulatory function of YOD1 for IL-1β signaling to NF-κB, reduction of YOD1 resulted in enhanced NF-κB target gene expression, which was especially evident at early stimulation time points. Taken together, overexpression and knock-down experiments suggest that YOD1 counteracts a rapid induction of NF-κB target genes in response to IL-1β stimulation.

To investigate if YOD1 is also controlling IL-1β responses in cells that mediate innate and inflammatory responses, we performed lentiviral shRNA transduction in murine immortalized bone marrow derived macrophages (iBMDM). Upon puromycin selection of shTRAF6- or shYOD1-transduced iBMDM, knock-down was verified by Western Blotting (*Figure 4G*). We monitored NF-κB signaling and activation (IκBα phosphorylation and degradation and NF-κB DNA binding) as well as target gene expression in TRAF6 or YOD1 knock-down iBMDM (*Figure 4G and H*). As expected, decreased TRAF6 expression severely reduced NF-κB activation and target gene expression upon IL-1β stimulation. In contrast, diminished expression of YOD1 augmented IκBα phosphorylation/ degradation as well as NF-κB DNA binding and enhanced the expression of TNFAIP3/A20 and NFKBIA/IκBα, revealing that YOD1 counteracts IL-1β triggered NF-κB signaling in iBMDM.

To determine the role of YOD1 in IL-1R induced signaling more precisely, we generated YOD1-deficient HeLa cells using CRISPR/Cas9 technology. For this, exon 4 of the YOD1 gene, which encodes almost the entire open reading frame, was deleted by transfection of flanking single guide RNAs together with Cas9 (*Figure 5—figure supplement 1A*). After clonal selection loss of YOD1 was analyzed by PCR of genomic DNA and Western Blotting (*Figure 5A*). The approach yielded two independent HeLa cell clones (#6 and #33) that carry the expected genomic deletion as verified by sequencing. Despite a faint residual PCR fragment at the size of YOD1 WT, no WT DNA could be

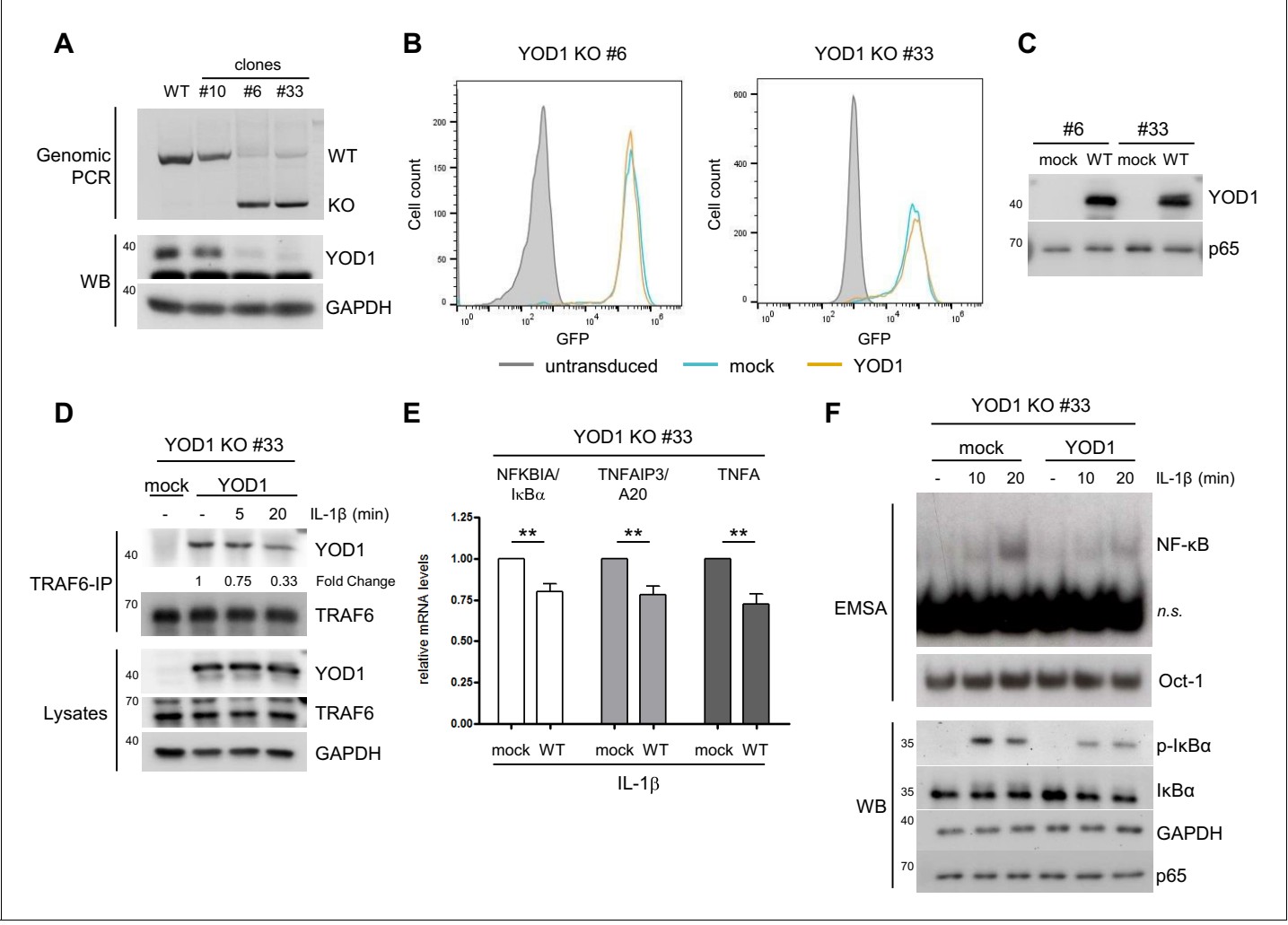

**Figure 5.** Reconstitution of YOD1-deficient HeLa cells impairs IL-1β-induced NF-κB signaling. (**A**) Validation of YOD1 KO HeLa cell clones. YOD1 genomic DNA and protein levels in parental HeLa cells and in cell clones generated by CRISPR/Cas9 gene editing were checked by PCR and Western Blot. (**B**) YOD1 deficient HeLa clones #6 and #33 are efficiently transduced with empty vector (mock) and YOD1 WT. Cells were transduced and homogenous populations of GFP expressing cells were sorted by FACS. FACS of GFP expression after sorting is shown. (**C**) Reconstitution of YOD1-deficient cell clones #6 and #33 with YOD1 WT. YOD1-deficient HeLa cells were transduced with YOD1 WT or mock constructs and YOD1 expression was analyzed by Western Blot. (**D**) YOD1/TRAF6 interaction in reconstituted YOD1-deficient HeLa clone #33 is decreasing upon IL-1R engagement. Cells were stimulated with IL-1β for the indicated time points. Anti-TRAF6 IPs were conducted and interaction of YOD1 was analyzed by Western Blot. Quantification of YOD1 bound to TRAF6 is shown. Numbers indicate the fold change after IL-1β stimulation (unstimulated set to 1). (**E**) Reconstitution of YOD1-deficient HeLa clone #33 with YOD1 WT diminishes NF-κB target gene expression. Cells were stimulated with IL-1β for 40 min. RNA was isolated and transcripts were analyzed by qRT-PCR as indicated. Bars show mean and SEM of seven independent experiments. Significance was evaluated using Student's t-test (*p<0,05; **p<0,01; ***p<0001; ns = not significant). (**F**) YOD1 re-expression in YOD1-deficient HeLa clone #33 diminishes NF-κB activation and IκBα phosphorylation and degradation. Cells were stimulated with IL-1β for the indicated time points and NF-κB DNA binding was assessed by EMSA (n.s. = non-specific band). Oct-1 EMSA served as loading control. IκBα phosphorylation and degradation was analyzed by Western Blot.

The following figure supplement is available for figure 5:

**Figure supplement 1.** Generation, reconstitution and analyses of YOD1-deficient HeLa cells.

detected by sequencing and the Western Blot demonstrates loss of YOD1 protein (*Figure 5A*). However, single cell clones from HeLa cells independent of the YOD1 status displayed a great heterogeneity with respect to cell proliferation, gene induction, NF-κB signaling etc. Therefore, we directly compared the effects within the individual YOD1 KO clones after lentiviral reconstitution, because

due to the high transduction efficiency clonal selection was not required. Cells were sorted by FACS to obtain homogenous population of GFP positive cells (*Figure 5B*) and expression of YOD1 was verified by Western Blot (*Figure 5C*). As observed earlier (*Figure 4B*), expression of YOD1 C160S was much weaker and therefore we focused the functional analyses on YOD1 WT reconstituted cells (*Figure 5—figure supplement 1B*). Co-IP revealed binding of reconstituted YOD1 to TRAF6 in unstimulated cells and the interaction was reduced after IL-1$\beta$ stimulation (*Figure 5D* and *Figure 5—figure supplement 1C*). On the level of NF-κB target gene expression we could verify the negative regulatory influence of YOD1 on gene induction as previously seen upon YOD1 overexpression or knock-down in HeLa cells (*Figure 5E*). Next, we examined direct effects on canonical NF-κB signaling in YOD1 KO clones in the absence (mock) or the presence of YOD1 (*Figure 5F* and *Figure 5—figure supplement 1D*). As expected for a putative negative regulator that controls TRAF6-dependent upstream signaling, activation of NF-κB DNA binding in response to IL-1$\beta$ stimulation was reduced in YOD1 expressing HeLa cells. In line, phosphorylation and degradation of the NF-κB inhibitor IκBα was reduced in YOD1 expressing cells (*Figure 5F* and *Figure 5—figure supplement 1D*). Thus, reconstitution of two independent KO cell clones provided clear evidence that YOD1 counteracts NF-κB signaling upon IL-1$\beta$ stimulation.

To directly compare the effects of the new negative IL-1 signaling regulator YOD1 with the positive regulators TRAF6 and p62 (*Sanz et al., 2000*; *Zotti et al., 2014*), we utilized siRNA based knock-down in HeLa cells. All three siRNAs yielded an efficient knock-down of their respective target on protein level (*Figure 6A*). Whereas knock-down of TRAF6 or p62 severely impaired IL-1$\beta$ triggered NF-κB activation as evident by EMSA, depletion of YOD1 enhanced NF-κB activation (*Figure 6A*). Congruently, induction of NF-κB target genes *NFKBIA/IκBα* and *TNFAIP3/A20* was decreased by TRAF6 or p62 knock-down and increased by YOD1 knock-down (*Figure 6B* and *Figure 6—figure supplement 1A*). Downregulation of TRAF6 or p62 prevented IκBα degradation and YOD1 depletion enhanced IκBα removal upon IL-1$\beta$ stimulation, demonstrating that all proteins affect NF-κB signaling (*Figure 6C*). Since TRAF6 acts upstream of IKK on the route to NF-κB, we verified that YOD1 also controls IKK activation by showing that YOD1 reduction coincides with increased IKK T-loop phosphorylation after IL-1$\beta$ treatment (*Figure 6D*). Of note, while IKK/NF-κB activation was significantly enhanced in the absence of YOD1, we found no effect of YOD1 knock-down on the activation of the MAPKs JNK, p38 and ERK (*Figure 6E*). TRAF6 is involved in NF-κB activation in response to IL-1R stimulation, but dispensable for TNFR triggered NF-κB activation (*Lomaga et al., 1999*; *Cao et al., 1996*). Since we did not observe an interaction between YOD1 and TRAF2, which is involved in TNFR signaling (*Hsu et al., 1996*; *Tada et al., 2001*), we directly compared the effect of YOD1 knock-down on NF-κB activation in response to either IL-1$\beta$ or TNFα stimulation (*Figure 6F*). While both stimuli induced NF-κB activation as evident by IκBα degradation and NF-κB DNA binding, YOD1 depletion led to strongly augmented NF-κB signaling and activation after IL-1R engagement, while TNFα-induced NF-κB signaling was unaffected by altered YOD1 amounts.

Absence or mutation of p62 has been shown to impede CD40- or RANK-induced signaling to NF-κB via TRAF6 in murine macrophages or osteoclasts, respectively (*Durán et al., 2004*; *Seibold and Ehrenschwender, 2015*). Thus, to determine a putative role of YOD1 in other TRAF6/p62-dependent pathways, we knocked-down TRAF6, p62 or YOD1 in CD40 expressing 293 or PC3 cells prior to stimulation with CD40-L or RANKL, respectively (*Figure 6—figure supplement 1B and C*). As expected, TRAF6 knock-down led to diminished IκBα degradation and NF-κB activation after CD40 or RANK stimulation. However, despite an efficient downregulation, p62 knock-down did not significantly affect NF-κB activation in response to both stimuli. Indeed, the previous studies already indicated that the role of p62 for CD40 and RANK signaling is relying on the cell type and the timing of stimulation (see Discussion) (*Durán et al., 2004*; *Seibold and Ehrenschwender, 2015*). In contrast to its negative regulatory role for IL-1 signaling, YOD1 downregulation impaired IκBα degradation and NF-κB activation upon CD40 or RANK stimulation in these cellular systems. Thus, the results indicate that YOD1 is not counteracting NF-κB signaling in general, but its negative function is restricted to the TRAF6/p62-dependent IL-1 pathway. In fact, in the case of CD40 and RANK signaling YOD1 even promotes NF-κB activation.

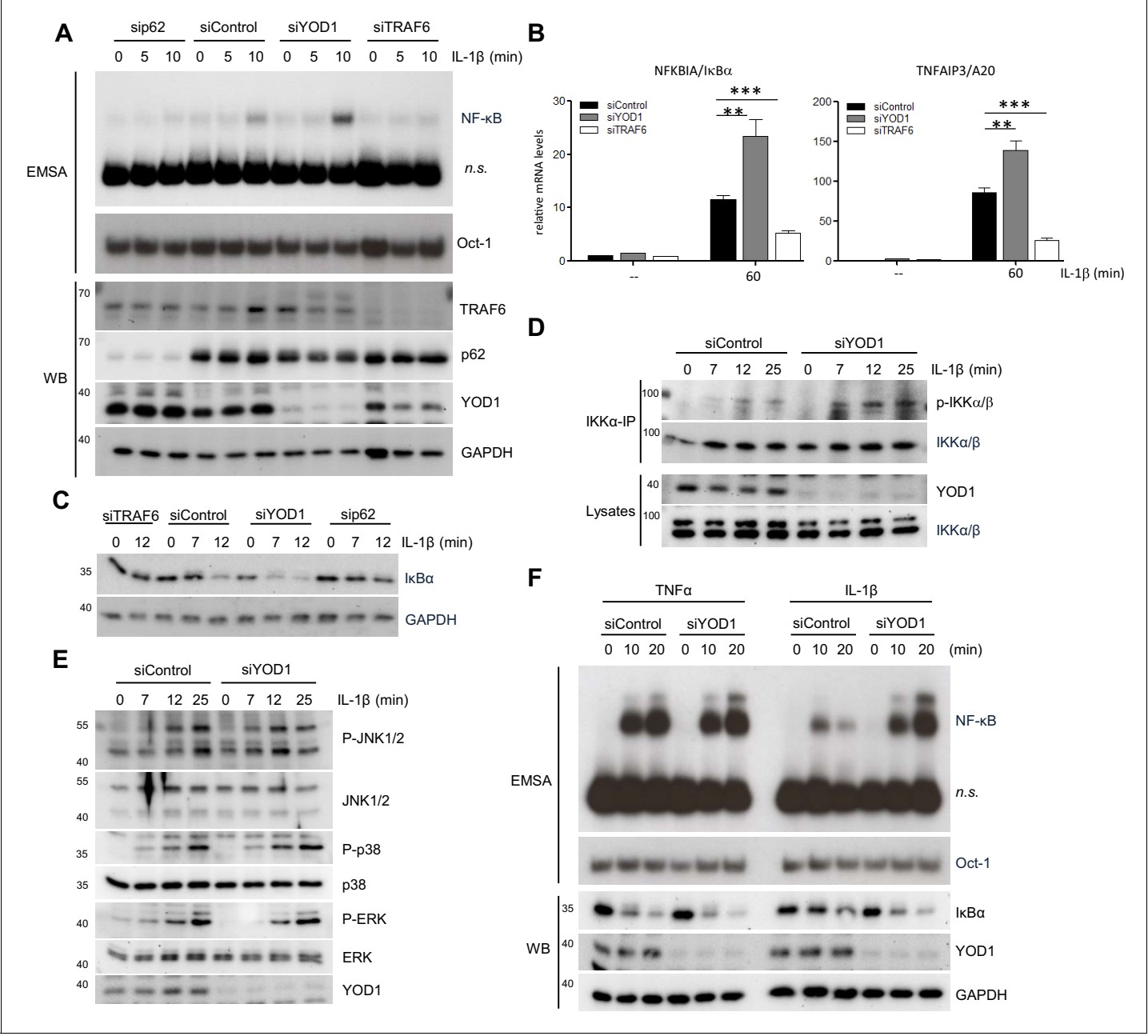

**Figure 6.** YOD1 and TRAF6/p62 exert opposing effects on IL-1β-induced NF-κB activation. (A) YOD1 knock-down promotes while TRAF6 and p62 knock-down impair NF-κB activation. HeLa cells were transfected with control siRNA or siRNA targeting YOD1, TRAF6 or p62 and stimulated with IL-1β as indicated. NF-κB and Oct-1 (control) DNA binding was assessed by EMSA (n.s. = non-specific band). Knock-down efficiency was confirmed by Western Blotting. (B) YOD1 knock-down promotes, while TRAF6 depletion impairs NF-κB target gene expression. HeLa cells were transfected with siRNA as indicated and subsequently stimulated with IL-1β for 60 min. Transcript levels were analyzed by qRT-PCR as indicated. Bars show mean and SEM of four to five independent experiments. Significance was evaluated using Student's t-test (**p<0,01; ***p<0001). (C) IκBα degradation is enhanced after YOD1 knock-down, but inhibited after TRAF6 or p62 knock-down. HeLa cells were transfected with siRNAs and stimulated with IL-1β as indicated. IκBα degradation was analyzed by Western Blot. (D) YOD1 is a negative regulator of IKK T-loop phosphorylation upon IL-1R engagement. HeLa cells were transfected with siControl or siYOD1 and stimulated with IL-1β as indicated. Anti-IKKα-IP was carried out to precipitate IKKα and IKKβ, and IKKα/β phosphorylation was analyzed by Western Blot. (E) YOD1 knock-down does not affect MAPK activation. HeLa cells were transfected with siRNA as in (D) and stimulated with IL-1β for the indicated time points. After cell lysis, MAPK activation was determined by Western Blotting using phospho-specific antibodies. (F) YOD1 specifically regulates IL-1β-, but not TNFα-induced NF-κB signaling. HeLa cells were transfected with siRNA as in (D) and stimulated with IL-1β or TNFα as indicated. NF-κB and Oct-1 (control) DNA binding was assessed by EMSA (n.s. = non-specific band). IκBα degradation and knock-down efficiency was confirmed by Western Blot.

*Figure 6 continued on next page*

*Figure 6 continued*

The following figure supplement is available for figure 6:

**Figure supplement 1.** Functional impact of TRAF6, p62 and YOD1 on CD40 and RANK stimulation.

## YOD1 counteracts TRAF6/p62-triggered ubiquitination in response to IL-1

To elucidate the mechanism how YOD1 counterbalances TRAF6- and p62-dependent IL-1 signaling, we determined the effect on ubiquitination events catalyzed by TRAF6. TRAF6 is an E3 ligase which in conjunction with the E2 enzyme UBC13/UEV1A transfers K63-linked ubiquitin chains onto its substrates (*Deng et al., 2000*; *Yin et al., 2009*). Also TRAF6 itself is ubiquitinated by an autocatalytic mechanism (*Lamothe et al., 2007*; *Wang et al., 2010*). By MYC-TRAF6 and GFP-YOD1 overexpression in HEK293 cells, we analyzed if YOD1 as deubiquitinating enzyme was able to remove ubiquitin chains conjugated to TRAF6 (*Figure 7A*). Under denaturing conditions (1% SDS) and after MYC-IP, TRAF6 ubiquitination is readily detectable and YOD1 expression did not interfere with TRAF6 poly-ubiquitination, which is in line with its previously reported inability to cleave K63-linked ubiquitin chains (*Mevissen et al., 2013*) (*Figure 7—figure supplement 1A*). Binding of p62 has been shown to enhance TRAF6 auto-ubiquitination (*Wooten et al., 2005*) and as expected, TRAF6 ubiquitination was strongly increased in the presence of p62 (*Figure 7A*). Noticeably, co-expression of YOD1 abrogated this enhancement. Also, catalytically inactive YOD1 C160S impaired the boost of TRAF6 auto-ubiquitination by p62, even though the inhibition was not quite as severe as with YOD1 WT. To validate that YOD1 does not directly cleave off ubiquitin chains from TRAF6, we incubated the p62-boosted TRAF6 ubiquitination with a panel of purified DUBs (*Figure 7—figure supplement 1B*). As expected, the non-selective DUB USP2 eradicated all TRAF6 ubiquitin modifications. The severe reduction of K63-linked and overall ubiquitination by the K63-specific DUB AMSH indicates predominant K63 ubiquitination of TRAF6. In contrast, neither recombinant YOD1 (K11, K27, K29 and K33 selectivity) nor Cezanne (K11 selectivity) were cleaving TRAF6 ubiquitin chains. Therefore, the data show that p62-induced attachment of K63-linked ubiquitin chains to TRAF6 is counterbalanced by YOD1 primarily by a non-catalytic mechanism that involves competition with p62 for TRAF6 association (compare *Figure 3E and F*).

To demonstrate that YOD1 also counteracts TRAF6 E3 ligase activity upon IL-1$\beta$ stimulation, we first determined inducible TRAF6 ubiquitination in YOD1 KO cells upon reintroduction of YOD1. TRAF6 was ubiquitinated upon IL-1$\beta$ treatment in mock transduced YOD1 KO HeLa cell clones, but TRAF6 ubiquitination was diminished upon re-expression of YOD1 (*Figure 7B* and *Figure 7—figure supplement 1C*). In fact, IL-1-induced TRAF6 ubiquitination correlates with a decreased binding of YOD1 to TRAF6 after IL-1 stimulation (see *Figure 5D* and *Figure 5—figure supplement 1C*), and we asked, whether reduced binding is also visible at the level of endogenous proteins in HeLa cells and primary HUVEC cells (*Figure 7C and D*). As noted earlier, YOD1 was co-precipitating with TRAF6 in unstimulated cells and the interaction was lost within the first 15–20 min of IL-1$\beta$ stimulation, revealing that TRAF6 is released from YOD1. To test whether endogenous p62 and YOD1 exert opposing functions in the regulation of TRAF6 ubiquitination in response to IL-1$\beta$, we knocked-down both proteins by siRNA and determined TRAF6 auto-ubiquitination (*Figure 7E*). Whereas p62 depletion completely abolished stimulus-dependent TRAF6 ubiquitination, down-regulation of YOD1 had an opposite effect, leading to increased ubiquitination after IL-1$\beta$ treatment. Again, to obtain data on the type of ubiquitin chains that TRAF6 is decorated with, we precipitated TRAF6 from IL-1$\beta$ stimulated HeLa cells and incubated it with a panel of DUBs (*Figure 7—figure supplement 1D*). As already seen with TRAF6 overexpression (*Figure 7—figure supplement 1B*), the promiscuous DUB USP2 and the K63 selective DUBs AMSH or, to a minor extent, TRABID cleaved TRAF6-attached ubiquitin chains. However, none of the other DUBs including YOD1 was able to remove TRAF6 ubiquitin chains, strongly suggesting that TRAF6 is primarily modified with K63-linked ubiquitin chains and that YOD1 is counteracting TRAF6 auto-ubiquitination by a non-catalytic mechanism. Ubiquitination of the IKK regulatory subunit NEMO is an important step in stimulus-dependent IKK activation and it was shown that NEMO ubiquitination induced by TRAF6 expression or IL-1$\beta$ stimulation

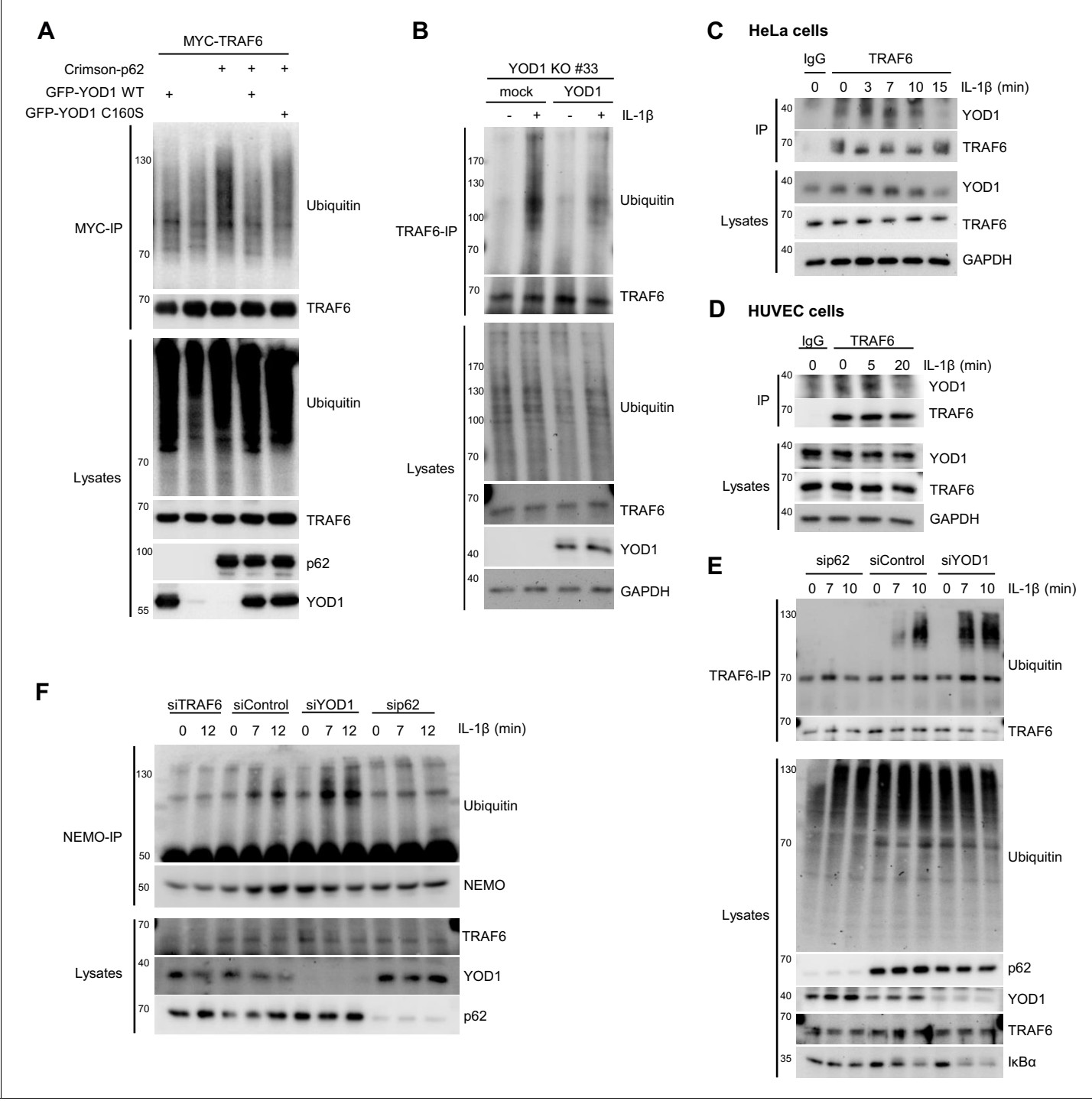

**Figure 7.** YOD1 counteracts TRAF6/p62-triggered ubiquitination. (A) YOD1 prevents augmented TRAF6 ubiquitination upon p62 binding. MYC-TRAF6, Crimson-p62, GFP-YOD1 WT and GFP-YOD1 C160S were co-transfected in HEK293 cells as indicated. After cell lysis under denaturing conditions (1% SDS), anti-MYC IP was conducted. TRAF6 ubiquitination was analyzed by Western Blot. (B) Reconstitution of YOD1-deficient HeLa cells diminishes TRAF6 ubiquitination. Mock or YOD1 reconstituted HeLa clone#33 was stimulated with IL-1β for 8 min and TRAF6 ubiquitination was analyzed as in (A), using anti-TRAF6 antibodies for IP. (C and D) Endogenous YOD1/TRAF6 interaction is lost upon IL-1β stimulation. HeLa cells (C) and HUVEC (D) were stimulated with IL-1β for the indicated time points and co-IPs were conducted using anti-TRAF6 or IgG antibodies. Co-IP of YOD1 was analyzed by Western Blot. (E) YOD1 knock-down promotes while p62 depletion inhibits IL-1-induced TRAF6 ubiquitination. HeLa cells were transfected with siRNAs and stimulated with IL-1β as indicated. TRAF6 ubiquitination was analyzed as in (B). (F) YOD1 knock-down promotes while TRAF6 and p62 knock-down impede NEMO ubiquitination. Experiment was essentially conducted as in (E), using anti-NEMO antibodies for IP.

*Figure 7 continued on next page*

*Figure 7 continued*

The following figure supplement is available for figure 7:

**Figure supplement 1.** TRAF6 poly-ubiquitination mainly consists of YOD1-resistant K63 linkages.

requires p62 (*Zotti et al., 2014*). Also in HeLa cells ubiquitination of NEMO after IL-1$\beta$ treatment was abolished in TRAF6 or p62 knock-down cells and YOD1 depletion had the opposite effect by enhancing stimulus-dependent NEMO ubiquitination (*Figure 7F*). Thus, YOD1 counteracts IL-1 signaling to NF-κB by functioning as a negative regulator of TRAF6/p62-mediated ubiquitination events.

## Discussion

The E3 ligase TRAF6 is involved in signaling in response to several NF-κB inducers (*Walsh et al., 2015*). Binding of the adapter p62/SQSTM1 enhances TRAF6 E3 ligase activity to promote NF-κB signaling upon IL-1 stimulation (*Sanz et al., 2000*; *Wooten et al., 2005*; *Durán et al., 2004*; *Seibold and Ehrenschwender, 2015*; *Cao et al., 1996*) Here, we identified the deubiquitinating enzyme YOD1 as a new regulator of TRAF6/p62-dependent IL-1R signaling to NF-κB. YOD1 is an OTU domain DUB that can hydrolyze K11, K27, K29 and K33 ubiquitin linkages (*Mevissen et al., 2013*). Structure-function analyses showed a high preference of the YOD1 OTU for K11- and K33- linked ubiquitin (*Flierman et al., 2016*) and as expected, we do not detect cleavage of K63-linked ubiquitin chains generated by TRAF6. A number of findings suggest that YOD1 controls TRAF6/p62- dependent IL-1 signaling predominantly by a non-catalytic mechanism: YOD1 as well as catalytically inactive YOD1 (i) compete with p62 for TRAF6 binding, (ii) abolish the formation of cellular p62/ TRAF6 aggregates, (iii) prevent enhancement of TRAF6 ubiquitination by p62 and (iv) inhibit IL-1- induced NF-κB activation upon overexpression. Since we see a slightly stronger reduction of p62- enhanced TRAF6 ubiquitination using YOD1 WT compared to catalytically inactive YOD1 C160S, it remains possible that YOD1 DUB activity can contribute to the negative regulation. Of note, in conjunction with the E2 enzyme UBC13/UEV1A TRAF6 catalyzes attachment of K63-linked chains (*Deng et al., 2000*; *Wang et al., 2001*), but TRAF6 and UBCH5A may build chains of different topology, including YOD1-sensitive K11-linked ubiquitin chains (*Windheim et al., 2008*; *Bosanac et al., 2011*). Further, K11-linked ubiquitin chains have been shown to be able to recruit NEMO and activate the IKK complex (*Dynek et al., 2010*), but relevant K11-modified substrates in the IL-1 pathway have not yet been defined. Using linkage specific OTU DUBs, we do not see formation of K11 ubiquitin chains on TRAF6, but YOD1 DUB activity may also control TRAF6 ubiquitination indirectly, e.g. by changing overall ubiquitin attachment. Nevertheless, we provide evidence that YOD1 acts in a non-catalytic competitive manner to counteract TRAF6 activation by p62.

Besides YOD1, the DUBs CYLD and A20 have been shown to control TRAF6 activity. Importantly, all three DUBs are interfering with TRAF6 activity via distinct mechanisms and are thus controlling different steps of an NF-κB response. As we described here for YOD1, CYLD is acting on TRAF6/ p62 complexes, but - in contrast to YOD1 - CYLD is not preventing the formation of TRAF6/p62 aggregates, but is recruited to TRAF6 by p62 (*Jin et al., 2008*; *Wooten et al., 2008*). Upon recruitment, CYLD is hydrolyzing K63-linked ubiquitin chains generated by active TRAF6 (*Wooten et al., 2008*; *Yoshida et al., 2005*; *Jin et al., 2008*). Similar to YOD1, A20 is not able to efficiently cleave K63 ubiquitin linkages and DUB activity is not required for impeding TRAF6 activity (*Mevissen et al., 2013*; *Shembade et al., 2010*). However, whereas YOD1 binds to TRAF6 in resting cells affecting C-terminal substrate binding, A20 is associating with TRAF6 only upon prolonged IL-1 stimulation to counteract binding of the E2 enzyme UBC13 to the RING-Z1 of TRAF6 (*Shembade et al., 2010*). Thus, CYLD and A20 act as negative feedback regulators that terminate post-inductive TRAF6 activity by a catalytic or non-catalytic mechanism, respectively. The YOD1/TRAF6 association in uninduced cells and the dissociation upon IL-1 stimulation indicate that YOD1 is acting in an earlier phase of the IL-1 response to counteract the accessibility of p62. In line, we show that early NF-κB signaling and gene induction is increased upon YOD1 depletion. Since p62 exerts a dual role by first activating TRAF6 and later recruiting CYLD (*Sanz et al., 2000*; *Jin et al., 2008*), CYLD may at least

partially impede enhanced signaling upon loss of YOD1 during an IL-1 response. Thus, our data together with the published data on CYLD and A20 reveal that the DUBs may act in a concerted manner at different steps of the pathway and that an interdependency of these negative regulators can potentially act as a fail-safe mechanism that can compensate for the loss of one another.

YOD1 depletion had no significant influence on IL-1-induced MAPK activation, even though TRAF6 is controlling p38 and JNK activation upon IL-1R engagement (*Lamothe et al., 2008*; *Ortis et al., 2012*). However, despite its role in NF-κB signaling, p62 is not substantially involved in the activation of JNK by TRAF (*Sanz et al., 2000*; *Feng and Longmore, 2005*). Thus, normal JNK signaling in YOD1 knock-down cells further supports the notion that YOD1 is selectively acting on p62/TRAF6 complexes. Hence, TRAF6 activation of MAPK and NF-κB signaling seems to involve different subsets of TRAF6 interactors.

The TRAF6/p62 signaling axis was shown to also mediate NF-κB activation in response to other inducers, including CD40, RANK or NGF stimulation (*Wooten et al., 2005*; *Durán et al., 2004*; *Seibold and Ehrenschwender, 2015*). Quite unexpectedly, we did not observe enhanced NF-κB signaling after YOD1 knock-down upon CD40 or RANK stimulation in 293 or PC3 cells. In contrast, NF-κB activation was even impaired upon decreased YOD1 expression. However, in this cellular context p62 knock-down did not significantly influence CD40 and RANK signaling. Previously, Seibold et al reported that CD40-induced NF-κB signaling was weakend in macrophages from p62-mutant mice, but unaffected in a human kidney tumor cell line after p62 knock-down (*Seibold and Ehrenschwender, 2015*). Further, p62 was shown to be dispensable for immediate early NF-κB signaling in response to RANK and it only controlled a second wave of NF-κB activity in osteoclasts after several days of stimulation (*Durán et al., 2004*). Apparently, the results reveal a stimulation, cell-type and context-dependent role of p62 for different TRAF6-dependent signaling pathways, which may explain why YOD1 is not acting as a general negative regulator in all these settings. However, why and how YOD1 promotes canonical NF-κB signaling and activation in the context of p62-independent CD40 or RANK stimulation is currently unclear and we can only speculate. Of note, YOD1 was originally identified as a co-factor of p97 in the regulation of protein quality control and the ERAD pathway (*Rumpf and Jentsch, 2006*; *Ernst et al., 2009*). YOD1/TRAF6 binding is apparently independent of p97, suggesting that YOD1 affects NF-κB signaling in response to IL-1 independent of this multifunctional AAA-ATPase. However, p97 was shown to positively regulate canonical NF-κB signaling by facilitating proteasomal IκBα degradation (*Li et al., 2014*; *Schweitzer et al., 2016*). Potentially, YOD1 could also function as a p97 co-factor for IκBα degradation to support canonical NF-κB signaling, but it is unclear why this would affect signaling in response to some inducers (e.g. CD40) while others are unaffected (e.g. IL-1 and TNF). Future studies will need to address if a putative positive action of YOD1 is relying on TRAF6 and/or p97 and in how far this regulatory events are cell-type and context dependent.

Clearly, YOD1 deficiency alone is not sufficient to induce TRAF6 ubiquitination or IKK/NF-κB signaling in the absence of any stimulation. However, changes in TRAF6 expression and TRAF6 oligomerization can activate downstream signaling (*Cao et al., 1996*; *Baud et al., 1999*). The C-terminal MATH of TRAF6 is an oligomerization and interaction domain (*Arch et al., 1998*; *Ha et al., 2009*) and stimulus-dependent recruitment of many adaptors including p62 is mediated by a consensus TRAF6 interaction motif (TIM) (*Ye et al., 2002*; *Linares et al., 2013*). The putative TIM of YOD1 is not required for binding to the TRAF6 MATH domain, indicating a different binding mode of YOD1 in un-induced cells. Nevertheless, YOD1 specifically binds to TRAF6 and not to TRAF2, and consequently YOD1 counteracts TRAF6-dependent NF-κB signaling from the IL-1R and not the TNFR. Since YOD1/TRAF6 interaction takes place in unstimulated cells, TRAF6 oligomerization is apparently not required, indicating that monomeric TRAF6 has a preference for binding to YOD1 over p62. Whether this is due to higher affinity or localization inside the cell needs to be elaborated, but the data suggest that YOD1 raises the threshold for TRAF6/p62 signaling to occur. Upon IL-1 stimulation, TRAF6 oligomerization may induce YOD1 dissociation, but also other processes like post-translational modification (e.g. TRAF6 ubiquitination) may play a role. The C-terminal MATH domain of TRAF6 associates with TIMs in various adaptors to regulate signaling in different settings. These adapters include MALT1 and Caspase8 in activated T cells (*Sun et al., 2004*; *Oeckinghaus et al., 2007*; *Bidère et al., 2006*), TIFA after IL-1 stimulation (*Takatsuna et al., 2003*; *Ea et al., 2004*) or TRIP6 in lysophosphatidic acid (LPA) stimulated cells (*Lin et al., 2016*). The existence of the large number of adapters that enhance TRAF6 activity and signaling underscores the necessity for tight

control and it will be interesting to analyze if YOD1 is influencing the recruitment of other C-terminal TRAF6 interaction partners to control NF-κB signaling in different settings.

We find that upon co-expression, YOD1 and TRAF6 are localizing to cytoplasmic aggregates that are distinct to p62/TRAF6 aggregates, the so-called sequestosomes. TRAF6 recruitment to p62 sequestosomes is enhanced upon IL-1β stimulation (*Sanz et al., 2000*; *Wang et al., 2010*). Sequestosomes are hotspots for signal transduction activity and in addition they can contribute to proteasomal degradation by co-localizing with the proteasome (*Seibenhener et al., 2004*). For NF-κB signaling, these two functions have been proposed to occur in a sequential process related to the progression of stimulation (*Wang et al., 2010*). Hence, freshly formed sequestosomes constitute a microenvironment for signaling to boost NF-κB activation (*Seibenhener et al., 2004*; *Sanz et al., 2000*). In the course of prolonged stimulation, sequestosomes mature into proteasome-organizing centers and NF-κB signaling is terminated (*Wang et al., 2010*). Our data indicate that YOD1 is able to counteract TRAF6 recruitment to sequestosomes and NF-κB signaling. Future analyses must elucidate the exact order of events and how multiple positive and negative regulators contribute to faithful initiation, maintenance and termination of sequestosome-mediated IL-1 signaling to NF-κB.

## Materials and methods

### Antibodies, siRNAs, shRNA and DNA constructs

The following antibodies were used: HA (clone 12CA5 (IP) and 3F1 (WB), obtained from E. Kremmer), IKKα (RRID: AB_396452 (IP)), NEMO (RRID:AB_398832), p62 (RRID:AB_398152 (WB)) (all BD Biosciences); ERK1/2 (RRID:AB_2141135, Calbiochem); p65 (RRID:AB_632037), Gal4-TA AD (RRID:AB_669111), IKKα/β (RRID:AB_675667 (WB)), MYC (RRID:AB_627268), NEMO (RRID:AB_2124846), TRAF6 (RRID:AB_793346 (IP)), p38 (RRID:AB_632138), p97 (RRID:AB_1568840), Ubiquitin (RRID:AB_628423) (all Santa Cruz Biotechnology); p-ERK1/2 (RRID:AB_331646), GAPDH-HRP (RRID:AB_1642205), IκBα (RRID:AB_10693636), p-IκBα (RRID:AB_10693636), p-IKKα/β (RRID:AB_331624), JNK1/2 (RRID:AB_2250373), p-JNK1/2 (RRID:AB_2307321), p-p38 (RRID:AB_331641), p97 (RRID:AB_2214632) (all Cell Signaling); p62 (RRID:AB_945626 (IP and WB)), TRAF6 (RRID:AB_778572 (WB)) (all Abcam); FLAG-M2 (RRID:AB_259529), GST (RRID:AB_259845), IgG rabbit (RRID:AB_1163661) YOD1 (RRID:AB_10600994 (WB) and RRID:AB_10599854 (IP or WB)) (all Sigma-Aldrich); Ubiquitin K63 (RRID:AB_1587580, Millipore); StrepTag II (RRID:AB_513133), HIS-Probe HRP (Thermo Scientific). The following siRNAs were used: siRNA pGL2 luciferase control, siYOD1: GGGAGGAGCAATAGAGATA, siTRAF6: GTTCATAGTTTGAGCGTTA, sip62: GGAAATGGGTCCACCAGGA (all Eurogentec); ON-TARGET*plus* non-targeting pool and ON-TARGET*plus* SMARTpool si-p97 (GE Dharmacon). shRNA sequence human shYOD1: GAGTACTGTGACTGGATCAAA, murine shYOD1: GCACAAATTGTAGCAAGTGAT, murine shTRAF6: ATCAACTGTTTCCCGACAATT; cDNAs were cloned into the following backbones: pcDNA3.1(+), pEF4HIS-C (Invitrogen), pGEX4T1 (GE Healthcare), pET28b+ (Novagen), pASK IBA 3+ (IBA Lifesciences), pFRED143 (*Ludwig et al., 1999*); pMD2.G, psPAX2, pLVTHM, pLV-tTRKRAB-red (*Wiznerowicz and Trono, 2003*), VSV-G, pLKO.1 (all obtained from Addgene); pSpCas9(BB)−2A-GFP (*Ran et al., 2013*) (PX458; Addgene); pGAD-C1 and pGBD-C1 (*James et al., 1996*).

### Cell culture, transfection and stimulation

HeLa (RRID: CVCL_0030), HEK-293 (RRID: CVCL_0045) and PC3 cells (RRID: CVCL_0035) were purchased from the DSMZ. U2OS cells (RRID: CVCL_0042) were purchased from ATCC. 293-CD40 cells (RRID: CVCL_9832) were a gift from Steve Ley and L929 (RRID: CVCL_0462) a gift of Andrea Oeckinghaus. Stocks from purchased and obtained cell lines were frozen after maximum of three passages and re-thawed every four to six weeks. Negative mycoplasma status of all cell lines was verified on a regular basis using a PCR testkit (A3744, Applichem) according to the manufacturer's protocol. Cells were grown in RPMI (PC3) or DMEM (all others) medium supplemented with 10% fetal calf serum (FCS) and 100 U/ml penicillin/streptomycin. 293 CD40 cells were grown and verified by Geneticin selection (*Coope et al., 2002*). Pools of primary HUVEC were purchased from Thermo Fisher Scientific and grown in Medium 200 supplemented with low serum growth supplement (Thermo Fisher Scientific). Experiments using HUVEC were carried out after a maximum of six passages. Murine BMDMs immortalized using J2 viurs (iBMDM) (*Gandino and Varesio, 1990*) were a

gift of Andrea Oeckinghaus and grown in DMEM medium supplemented with 10% FCS and conditioned medium (10–30% L929 cell supernatant).

HEK293 cells were transfected using standard calcium phosphate precipitation protocols. U2OS and HeLa cells were transfected using Lipofectamine LTX and 3000 according to the manufacturer´s protocol (Thermo Fisher Scientific). For RNA interference, HEK293, 293 CD40 and HeLa cells were transfected with 100 nM siRNA and Atufect transfection reagent (1,0 µg/ml) (Silence Therapeutics) and analyzed after 72 hr. HeLa and HUVE cells were stimulated with human IL-1$\beta$ (R and D systems) in concentrations ranging from 0,5 ng/ml (qRT-PCR and EMSA) to 5 ng/ml (endogenous co-IPs) or with human TNF$\alpha$ (Biomol; 5 ng/ml). 293-CD40 were stimulated with 0,25 µg/ml CD40 ligand (Source Bioscience). For stimulation with recombinant RANK-L (R and D systems, 150 ng/ml), PC3 cells were serum starved overnight. iBMDM were stimulated with murine IL-1ß (PeproTech; 2 ng/ml).

## Lentiviral transduction

For inducible YOD1 expression or shRNA knock-down, HeLa cells were double-infected with lentiviruses to generate a DOX-inducible expression system based on tTR-KRAB hybrid protein (*Wiznerowicz and Trono, 2003*). Cells were first infected with pLV-tTRKRAB-red vector (IRES exchanged for T2A) and afterwards with pLVTHM-based transfer vectors encoding YOD1 WT, YOD1 C160S or shYOD1 sequence, respectively, plus GFP as marker. For constitutive YOD1 expression, a single infection round with the transfer vector encoding YOD1 WT or empty transfer vector (mock) was conducted. Lentivirus production and transduction was essentially performed as described previously (*Hadian et al., 2011*), using pMD2.G and psPAX2 as envelope and packaging plasmids. Virus was applied to HeLa cells for about 18 hr. To induce protein or shRNA expression, cells were treated with 0,05 µg/ml DOX (Roth) for 72 hr. Transduction efficiency was analyzed by flow cytometry with an Attune Acoustic Focusing Cytometer System (Thermo Scientific) on the basis of GFP or dsRed expression. qRT-PCR and Western Blotting were performed to determine mRNA and protein expression, respectively. If necessary, positively transduced cells were sorted by FACS to yield culture with more than 90% transduced cells.

For shRNA knock-down in murine iBMDM, cells were infected with lentiviral pLKO.1 vectors encoding shRNA constructs (shYOD1 and shTRAF6) or empty vector pLKO.1 (shMock), using VSV-G and psPAX2 as envelope and packaging plasmids. Virus was applied for about 30 hr and successfully transduced cells were selected by puromycin treatment (3 µg/ml). qRT-PCR and Western Blotting were performed to determine mRNA and protein expression, respectively.

## Recombinant protein expression, purification and GST pull-down

Recombinant proteins were expressed in *E. coli* BL21 RIPL codon plus (Agilent Technologies) and purified by affinity chromatography using an ÄKTA protein purification system (GE Healthcare). Purified proteins were taken up in storage buffer (PBS for YOD1 and p97; 20 mM Tris (pH 8), 20 mM NaCl, 1% Glycine, 0,5% Mannitol for HIS-TRAF6; 20 mM Tris (pH 8), 20 mM NaCl, 100 µM $ZnCl_2$, 1 mM DTT for Strep-TRAF6). For GST-PDs, Glutathione Sepharose 4B beads (GE Healthcare) were saturated with GST or GST-YOD1 for 1 hr at 4°C (assay buffer: PBS, 5% Glycerole, protease inhibitor cocktail (Roche) or 20 mM Tris (pH 8), 20 mM NaCl, 1% Glycine, 0,5% Mannitol, protease inhibitor cocktail), followed by extensive washing. Subsequently, the candidate interacting protein was incubated with the bead-bound GST-protein in assay buffer complemented with 0,5% Triton X100 for 2 hr at 4°C. Again, beads were washed extensively. PDs were analyzed by SDS-PAGE and Coomassie Staining or Western Blotting.

## Yeast-two-hybrid

Competent *S. cerevisiae* were prepared using a standard protocol (*Knop et al., 1999*). Proteins of interest were fused to GAL4 transcription factor activation domain (AD) or binding domain (BD) using pGAD-C1 and pGBD-C1 vectors, respectively. AD and BD plasmids contained LEU and TRP as markers, respectively. Expression constructs were transformed in PJ69-7A cells and spotted on -LEU-TRP selection media (+HIS) to monitor successful co-transformation and on -HIS-LEU-TRP selection media (−HIS) to monitor protein-protein interaction.

### Generation of YOD1-deficient HeLa cells by CRISPR/Cas9

Two sgRNAs targeting regions flanking exon 4 (5'- AGCATAAACTGGGGTTACTA −3' and 5'- TTAGGGTTACCATAGCTTAT −3') were cloned into px458-GFP vector containing Cas9 and GFP. HeLa cells were lipofected with sgRNA expressing plasmids. GFP positive cells were sorted by FACS and clonal cell lines were isolated by serial dilution. After expansion, cell clones were genotyped using PCR with intronic primers (see Appendix) flanking both sides of exon 4. PCR products were verified by sequencing. YOD1 protein expression was analyzed by Western Blot.

### Immunoprecipitation, western blot, electrophoretic mobility shift assay

Co-immunoprecipitations (IP) and Western Blotting were done as described (*Schimmack et al., 2014*; *Meininger et al., 2016*). For anti-TRAF6 IPs in HeLa cells, a CHAPS containing lysis buffer was used (40 mM HEPES (pH 7,4), 120 mM NaCl, 1 mM EDTA, 0,3% CHAPS, 0,5 M NaF, 1 M DTT, 1 M $\beta$-Glycerophosphate, 200 mM sodium vanadate, protease inhibitors). For electrophoretic mobility shift assay (EMSA), cells were lysed in whole cell lysis high salt buffer (20 mM HEPES (pH 7,9), 350 mM NaCl, 20% Glycerol, 1 mM $MgCl_2$, 0,5 mM EDTA, 0,1 mM EGTA, 1% NP-40, 0,5 M NaF, 1 M DTT, 1 M ß-Glycerophosphate, 200 mM sodium vanadate, protease inhibitor cocktail (Roche)). Equal amounts of extract were subjected to NF-κB and Oct-1 EMSA as described previously (*Meininger et al., 2016*).

Quantification of Western Blots was conducted using LabImage 1D Software (Kapelan Bio-Imaging). To quantify protein amounts after co-IP, YOD1 and p62 normalization to amounts in the lysates and to co-precipitated TRAF6 was performed to adjust for differences in transfection and precipitation efficiencies.

### Detection of cellular ubiquitination

To analyze cellular protein ubiquitination, cells were lyzed in co-IP buffer (150 mM NaCl, 25 mM HEPES (pH 7,5), 0,2% NP-40, 1 mM Glycerol, 0,5 M NaF, 1 M DTT, 1 M $\beta$-glycerophosphate, 200 mM sodium vanadate, protease inhibitor cocktail (Roche)) supplemented with 1% SDS. After repeated passing through a 26G-syringe, lysates were boiled at 95°C, cooled down on ice and centrifuged. For subsequent IP, supernatant was taken and SDS was diluted with co-IP buffer to a final SDS concentration of 0,1%. IP was carried out as described above.

### In vitro YOD1 cleavage and UbiCrest assays

To monitor enzymatic activity of recombinant GST-YOD1, 100 ng of the DUB was incubated with 250 ng recombinant tetra-ubiquitin chains in DUB buffer (50 mM Tris (pH 7,5)/0,03% BSA/5 mM DTT) at 37°C. Samples were taken at the indicated time points. Samples for time point zero were taken before adding the DUB. Cleavage efficiency was analyzed by Western Blotting. To analyze chain composition of TRAF6 poly-ubiquitination, over-expressed (HEK293 cells) or endogenous (HeLa cells) TRAF6 was precipitated by IP. Chain restriction analysis was performed using the UbiCREST Kit (Boston Biochem) (*Hospenthal et al., 2015*) according to the manufacturer's protocol. Samples were analyzed by Western Blotting and by Silver Staining of SDS-PAGE using Pierce Silver Stain Kit (Thermo Fisher Scientific).

### Confocal fluorescence microscopy, plot profiling and automated analyses of FI co-clustering

For intracellular protein localization studies, spinning disk confocal fluorescence microscopy was conducted in 96well plate format (View Plate Glass and Cell Carrier) using an Operetta high-content imaging system (all PerkinElmer). Cells were settled on poly-D-lysine coated plates and transfected as described above using Lipofectamine. Approximately 24 hr post transfection, cells were fixed in 2% PFA and cell nuclei were stained with Hoechst33342 (Life Technologies). Images were taken with a 60x objective and analyzed with Columbus Software (PerkinElmer). To quantify fluorescence intensities we performed plot profile analysis. Raw data of all channels were imported to ImageJ (RRID: SCR_003070) software. Pictures of single channels were re-merged and analyzed along the straight line indicated in the Figure with the 'Plot Multicolor 4.3' Plugin.

Multiparametric image analysis was performed using the Columbus Software 2.5 (PerkinElmer). To identify cells, nuclei were detected via the Hoechst signal. In cells transfected with GFP-YOD1

the complete cell was detected via basal GFP-signal. Background was determined as overall mean signal of the respective fluorescence in transfected cells. Each Crimson-/CFP-p62-, GFP-YOD1- or RFP-TRAF6 spot was automatically detected by the software as a small region within the corresponding image by having a higher intensity than its surrounding area. For quantitative analyses we determined the Crimson-/CFP-, GFP- and RFP-signal in the corresponding spots or control areas.

## Quantitative real-time PCR

Equal amounts of RNA (InviTrap Spin Universal RNA Mini Kit, 1060100200, Stratec) were transcribed into cDNA using Verso cDNA synthesis Kit (AB1453B, Thermo Fisher Scientific). Quantitative real-time (qRT) PCR was performed using KAPA SYBR FAST qPCR Master Mix (KAPA Biosystems) and standard LightCycler protocol on a Roche LightCycler 480. RNA-Polymerase II (RPII), and Hydroxy-methylbilane synthase (HMBS) and 18S rRNA served as internal standard. For primer sequences see Appendix.

## Data analysis

Each experiment shown in the paper represents at least two to three biological replicates with similar results. Statistical significance was determined by Student's t-test using GraphPad Prism5 software (RRID:SCR_002798) and sample size is mentioned for those experiments in the respective figure legend. Data are depicted as mean ± SEM.

## Acknowledgements

We thank Katrin Demski and Simon Widmann for excellent technical assistance. We thank Elisabeth Kremmer for gifting anti-HA antibodies, Steve Ley for providing CD40 293 cells and Andrea Oeckinghaus for providing iBMDM and helping with the lentiviral transduction protocol. The following vectors were kindly provided: PX458 by Feng Zhang (Addgene #48138); pMD2.G, psPAX2, pLVTHM, pLV-tTRKRAB-red (all Didier Trono; Addgene # 12259, 12260, 12247, 12250). Atufect lipofection reagent was a kind gift from Silence Therapeutics, Berlin.

## Additional information

### Funding

| Funder | Grant reference number | Author |
|---|---|---|
| Deutsche Forschungsgemeinschaft | SPP1365 | Daniel Krappmann |
| Wilhelm Sander-Stiftung | 2012.075.2 | Daniel Krappmann |
| Deutsche Forschungsgemeinschaft | SFB1054 A4 | Daniel Krappmann |

The funders had no role in study design, data collection and interpretation, or the decision to submit the work for publication.

### Author contributions

GS, Conceptualization, Formal analysis, Investigation, Visualization, Methodology, Writing—original draft, Writing—review and editing; KS, Formal analysis, Investigation, Visualization, Methodology; KK, Formal analysis, Investigation; TG, JKB, Investigation, Methodology; KH, Supervision, Funding acquisition, Investigation; DK, Conceptualization, Supervision, Funding acquisition, Investigation, Visualization, Methodology, Writing—original draft, Project administration, Writing—review and editing

### Author ORCIDs

Daniel Krappmann, http://orcid.org/0000-0001-7640-3234

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

## Appendix

### PCR primer

#### Human qPCR primers

| | |
|---|---|
| RPII fw | 5'-GCACCACGTCCAATGACA-3' |
| RPII rev | 5'- GTGCGGCTGCTTCCATAA-3' |
| HMBS fw | 5'-GCTGCAACGGCGGAA-3' |
| HMBS rev | 5'-CCTGTGGTGGACATAGCAATGATT-3' |
| 18S rRNA fw | 5'-GCTTAATTTGACTCAACACGGGA-3' |
| 18S rRNA rev | 5'-AGCTATCAATCTGTCAATCCTGTC-3' |
| NFKBIA (IκBα) fw | 5'-CCGCACCTCCACTCCATCC-3' |
| NFKBIA rev | 5'-ACATCAGCACCCAAGGACACC-3' |
| TNFAIP3 (A20) fw | 5'-TTTTGTACCCTTGGTGACCCTG-3' |
| TNFAIP3 rev | 5'-TTAGCTTCATCCAACTTTGCGG-3' |
| TNFA fw | 5'-cccagggacctctctctaatca-3' |
| TNFA rev | 5'-gctacaggcttgtcactcgg-3' |

#### Murine qPCR primers

| | |
|---|---|
| HMBS fw | 5'-GCGCTAACTGGTCTGTAGGG −3' |
| HMBS rev | 5'-TGAGGGAAAGGCAGATATGGAGG-3' |
| NFKBIA (IκBα) fw | 5'-TTGCTGAGGCACTTCTGAAAG-3' |
| NFKBIA rev | 5'-TCTGCGTCAAGACTGCTACACT −3' |
| TNFAIP3 (A20) fw | 5'-GCTCAACTGGTGTCGTGAAG-3' |
| TNFAIP3 rev | 5'-ATGAGGCAGTTTCCATCACC-3' |

#### Intronic primers for genomic PCR

| | |
|---|---|
| YOD1 fw | 5'- TTGTTTACTCCAGACCCCTTCACTAAATTGGGATGCAACC −3' |
| YOD1 rev | 5'- GTCAGTAGGTGGCAAGGATCACCTATTCTGTCACTCCAGC −3' |

