## [Decision Letter]

Thank you for submitting your article "YOD1/TRAF6 association balances p62-dependent IL-1 signaling to NF-κB" for consideration by *eLife*. Your article has been favorably evaluated by Jonathan Cooper (Senior Editor) and three reviewers, one of whom is a member of our Board of Reviewing Editors.

The reviewers have discussed the reviews with one another and the Reviewing Editor has drafted this decision to help you prepare a revised submission. We hope you will be able to submit the revised version within two months.

Essential revisions:

Thank you for submitting your study to publication in *eLife*. The three reviewers shared appreciation to the importance of your findings. However, they had several comments and requests. They felt that the fidelity of your conclusions requires substantiation by tests done in additional cells (including primary cells), and should utilize other triggers that activate TRAF6 dependently on p62. They also raised concerns about the quality of some of the data and about their modes of presentation.

I hope you will be able to address these concerns and requests satisfactorily, and look forward to your revised manuscript.

Reviewer #1:

This study shows that the deubiquitinating enzyme YOD1 can bind to the MATH domain of *TRAF2* and that this binding is competitive with that of the binding of p62/Sequestosome-1 to TRAF6. The latter binding has been shown to occur following stimulation of TRAF6 by a number of agents and to be required for triggering the ubiquitin ligase function of TRAF6. The current study further shows that in two cell lines – HeLa and HEK293, YOD1 associates constitutively with TRAF6 and dissociates from it in response to IL-1.

These findings are interesting. However, there is need to supplement them with additional basic information about their potential physiological significance.

In the Discussion section the authors state that since "expression of TRAF6 alone is sufficient to induce strong NF-B activation…TRAF6 activity needs to be tightly controlled by negative regulatory processes" and pursue to suggest that YOD1 mediates such negative regulation. However, as the authors themselves point out, their data are actually not consistent with such a role for YOD1 since neither knock down nor knockout of YOD1 results in spontaneous TRAF6 signaling.

As the authors point out, YOD1 also does not serve as a negative feedback inhibitor of IL-1 signaling. Although its over-ex-expression somewhat inhibits TRAF6 mediated NF-B activation by IL-1, this over-expression does not recapitulate any known physiological scenario since, unlike CYLD and A20, YOD1 is not known to be upregulated in response to IL-1.

Knockdown or knockout of YOD1 is shown to enhance the response to IL1. However, this change is also not known to recapitulate any physiological scenario.

While a final definition of the physiological implication of the findings might not be reachable at this stage, the authors should be requested to provide information on two subjects that are central to any future consideration of the potential answers:

1) To explore the possibility that modulation of TRAF6 signaling by variation in the cellular level of YOD1 is physiologically relevant, the authors should compare the cellular level of YOD1 to that of TRAF6 in several different types of cells and also assess the extent of their association in these different cells.

2) The need for p62 binding to TRAF6 in order to trigger signaling by the latter is not restricted to the IL-1 effect. Several other triggers that employ TRAF6, including mTOR, CD40, RANK and Nerve Growth Factor also require association of p62 with TRAF6. To confirm that the effect of YOD1 on TRAF6 activation indeed relates to its ability to interfere with p62 binding to the latter, rather than affecting some other aspect of IL-1 signaling, the authors should check if ablation of YOD1 expression facilitates signaling in response to such other triggers.

Reviewer #2:

In this study, Schimmack et al. identify the deubiquitinating enzyme YOD1 as a new interacting protein of the E3 ligase TRAF6. The authors show that YOD1 interacts with the C-terminal MATH domain of TRAF6, and inhibits TRAF6 binding with p62/Sequestosome-1. Functional studies using HeLa YOD1 overexpression and knockout cells support the idea that YOD1 is a negative regulator of IL-1-induced NF-κB activation. YOD1 appears to inhibit TRAF6 by a noncatalytic mechanism since the YOD1 C160S mutant still inhibits IL-1-NF-κB signaling. Overall, this is a very thorough and interesting study and the data are convincing for the most part. However, a weakness of the study that needs to be addressed is that much of the data were generated using overexpression strategies in cell lines (mainly HeLa and 293).

1) Almost all functional studies (stable overexpression of YOD1 and CRISPR knockouts) were performed in HeLa cells. Some of the key data should be confirmed in other cell lines, and potentially primary cells.

2) It is surprising that mutation of the putative TRAF6 interaction motif in YOD1 does not abolish the interactions between YOD1 and TRAF6. Have the authors examined if additional mutations (e.g. P94A) in YOD1 impair TRAF6 binding?

3) There is significant potential for artifacts with overexpression of fluorescently tagged proteins in Figure 2 and 3. The authors should attempt imaging studies with endogenous YOD1, p62 and TRAF6 proteins to confirm the key results.

4) The EMSAs in Figure 5 and Figure 6 could be improved by the inclusion of an Oct-1 probe as a control.

5) In Figure 7, the authors conclude from this data that YOD1 is released from TRAF6 after 15 min. of IL-1 stimulation. However, there is also less YOD1 expressed in the lysates at this time point. A loading control (e.g. β-actin) is needed here.

6) For experiments examining the ubiquitination of endogenous TRAF6 and NEMO (e.g. Figure 7), the authors should also blot with K63 linkage specific Ub antibody. The Ub blot for Figure 7—figure supplement 1 is of poor quality and should be repeated.

Reviewer #3:

The deubiquitinase YOD1 (OTUD2) was identified in a two-hybrid screen with TRAF6 and confirmed to interact by co-immunoprecipitation from lysates of unstimulated Hela cells. YOD1 was shown to compete for TRAF6 binding to p62/sequestosome-1, an interaction that is required for IL-1R signaling. Knockdown and overexpression of YOD1 suggested that YOD1 functions as a negative regulator of TRAF6/p62-dependent activation of NF-κB by IL-1β.

Given the importance of IL-1β in inflammation and immune responses, the regulation of IL-1R signaling is of considerable interest. The study by Schimmack et al. contains interesting new information on this subject. However, the quality of some of the results needs to be improved and quantification of some of the data plus statistical testing for significant differences is essential to support the proposed model.

Specific points:

1) To test the specificity of YOD1 interaction with TRAF6, a panel of E3 ligases and other Ub regulatory proteins was tested for interaction with YOD1 by yeast two-hybrid assay (Figure 1—figure supplement 1). Do the authors have evidence that the constructs used for this panel actually work in a yeast two-hybrid assay?

2) In Figure 1, HA-TRAF6 is shown to co-immunoprecipitate with FLAG-YOD1. However, the levels of HA-TRAF6 { ± } FLAG-YOD1 co-expression are very different, so it is difficult to assess whether this interaction is specific. This result should be replaced with another experiment in which HA-TRAF6 levels are more equivalent { ± } FLAG-YOD1.

3) In Figure 1, it is not clear whether cells have been transfected with empty vector (EV) in the third lane from the left. There appears to be a small amount of FLAG-YOD1 in the HA immunoprecipitates in this lane, although the cell lysates appear to have less FLAG-YOD1. Is the co-immunoprecipitation of FLAG-YOD1 specific?

4) The level of HA-TRAF6 expressed in the left-hand lane co-transfected with EV (Figure 1) is much lower than the lysates co-expressing FLAG-YOD1 proteins. Consequently, it is difficult to assess the specificity of HA-TRAF6 co-immunoprecipitation with FLAG-YOD1.

5) siRNA knockdown of YOD1 is used to show the specificity of endogenous YOD1 co-immunoprecipitation with TRAF6 (Figure 1). In reality, this experiment simply confirms the identity of the YOD1 and demonstrates that there is less YOD1 present in the cell lysates to co-immunoprecipitate with TRAF6. This experiment should be replaced with one in which endogenous TRAF6 is knocked down and the effect on YOD1 co-immunoprecipitation with anti-TRAF6 tested.

6) It would strengthen the paper if endogenous TRAF6 and YOD1 could be shown to associate in cell lysates from primary human cells (e.g. HUVECs).

7) The authors state that GFP-YOD1 and RFP-TRAF6 co-localize to a large extent when co-expressed (Figure 2). This conclusion would be stronger if the extent of co-localisation was quantified in multiple cells and tested for statistical significance.

8) How representative are the results shown in Figure 3? Imaging results should be shown from multiple cells. If possible imaging data should be quantified and colocalization/non-colocalization tested statistically.

9) The authors claim that YOD1 expression decreases the interaction of TRAF6 with p62 (Figure 3). However, the level of p62 in the lysates is reduced in cells co-expressing YOD1. To convincingly demonstrate that YOD1 prevents binding of p62 to TRAF6, p62 levels in lysates need to be similar { ± } YOD1 co-expression. In addition, data need to be quantified from multiple experiments and differences tested for statistical significance.

10) Based on microscopic examination of cells expressing RFP-TRAF6, Crimson-p62 and GFP-YOD1 (Figure 3), the authors conclude that p62 was excluded from YOD1/TRAF6 clusters. Only images of single cells are shown. How representative are these data (see point 8)?

11) Using HeLa cells that inducibly express YOD1 after doxycycline treatment, demonstrate that both WT and C160S YOD1 co-immunoprecipitate with TRAF6 (Figure 4). Control IgG immunoprecipitates are required to confirm the specificity of this interaction. The authors claim that IL-1β stimulation reduces YOD1-TRAF6 association. However, it appears that IL-1β stimulation actually increases association of WT YOD1 with TRAF6 while reducing that with YOD1[C160S]. The authors should comment on this difference and confirm reproducibility of these results by quantitation from multiple experiments and statistical testing.

12) Based on the results shown in Figure 4, the authors conclude that overexpression of WT or C160S YOD1 reduces IL-1β activation of three NF-κB-dependent genes. Consistent with this result, shRNA knockdown of YOD1 increases expression of the same three genes. Does doxycycline treatment alone (without YOD1 expression) alter IL-1β activation of NF-κB in the parental cells?

13) Figure 6 shows the inhibitory effect of TRAF6 knockdown on IL-1β induced expression of Nfkbia and Tnfaip3 mRNAs. Does p62 knockdown affect the expression of these genes?

---

## [Author Response]

Essential revisions:

Thank you for submitting your study to publication in eLife. The three reviewers shared appreciation to the importance of your findings. However, they had several comments and requests. They felt that the fidelity of your conclusions requires substantiation by tests done in additional cells (including primary cells), and should utilize other triggers that activate TRAF6 dependently on p62. They also raised concerns about the quality of some of the data and about their modes of presentation.

We thank the reviewers for their in-depth review and helpful suggestions. As suggested, we have performed more experiments and included additional data to support our findings. Most important, we performed shRNA-mediated knock-down in immortalized bone marrow derived macrophages (iBMDM) and show that while TRAF6 knock-down impairs, YOD1 enhances NF-κB signaling and gene expression in response to IL-1 (Figure 4). Further, we also show binding data in primary cells: YOD1 and TRAF6 interact in primary human umbilical vein endothelial cells (HUVEC) (Figure 1).

Moreover, just like in HeLa cells, TRAF6/YOD1 interactions declines after IL-1 stimulation in HUVEC cells (Figure 7), providing support for the relevance and regulation of this interaction.

Concerning the question of other triggers that rely on TRAF6 and p62, we determined the effect of siRNA knock-down of TRAF6, p62 and YOD1 upon CD40 signaling in CD40 overexpressing 293 cells and after RANKL stimulation in PC3 cells (Figure 6—figure supplement 1). Quite to our surprise, we found that in both settings TRAF6 but not p62 is required for optimal NF-κB activation. However, the requirement of p62 for initial TRAF6-dependent CD40 and RANK signaling is not so clear (see also below response to reviewer# 1; Duran et al., Dev Cell 2004; Seibold et al., JCMM, 2015). Further, YOD1 was not antagonizing NF-κB activation after CD40L and RANKL stimulation, but even necessary for NF-κB signaling. Thus, the data are pointing to a more complex interplay in which depending on the stimulus, YOD1 can act as a negative or a positive regulator of NF-κB and we included these important data. At the moment, we cannot resolve how YOD1 exerts these opposing effects. Of note, it is even possible that the positive effect is relying on YOD1 activity as a cofactor of p97/VCP, which was recently shown to enhance IκBα degradation. We changed the Discussion (third paragraph). However, we think that in this manuscript we need to keep the focus on IL-1R signaling that clearly relies on the interplay of TRAF6 and p62 (see also our response to reviewer#1). All experiments and changes have been described in more detail in the point by point response to the reviewer comments. We are confident that we have been able to adequately address the issues and to improve the manuscript.

I hope you will be able to address these concerns and requests satisfactorily, and look forward to your revised manuscript.

Reviewer #1:

[…] These findings are interesting. However, there is need to supplement them with additional basic information about their potential physiological significance.

*In the Discussion section the authors state that since "expression of TRAF6 alone is sufficient to induce strong NF-B activation…TRAF6 activity needs to be tightly controlled by negative regulatory processes" and pursue to suggest that YOD1 mediates such negative regulation. However, as the authors themselves point out, their data are actually not consistent with such a role for YOD1 since neither knock down nor knockout of YOD1 results in spontaneous TRAF6 signaling.*

Indeed, depletion of YOD1 alone is not sufficient to induce NF-κB signaling. We agree that the sentence (“Expression of TRAF6 alone …”) may be misleading. In fact, Cao et al. and others showed that overexpression of TRAF6 induces strong activation of NF-κB, but this is quite different from an endogenous setting and may not be controlled by YOD1. For clarity, we took out this sentence. The lack of spontaneous activation shows that YOD1 cannot be the sole regulator that antagonizes TRAF6 in uninduced cells. Nevertheless, the YOD1/TRAF6 binding in the absence of stimulation still suggests an involvement in maintaining inactive TRAF6 or antagonizing early TRAF6 activation and we think it is worthwhile to discuss this especially in relation to CYLD and A20.

As the authors point out, YOD1 also does not serve as a negative feedback inhibitor of IL-1 signaling. Although its over-ex-expression somewhat inhibits TRAF6 mediated NF-B activation by IL-1, this over-expression does not recapitulate any known physiological scenario since, unlike CYLD and A20, YOD1 is not known to be upregulated in response to IL-1.

We agree and we are not claiming that YOD1 is a negative feedback inhibitor. However, not all negative regulators of the pathway must induce and/or regulate negative feedback loops. A good example is OTULIN that functions in homeostatic control by antagonizing LUBAC E3 ligase activity (Keusekotten et al., Cell, 2013). We show here that YOD1 antagonizes TRAF6/p62 complexes and we also modified the wording at the end of the Abstract to avoid any misunderstanding.

Knockdown or knockout of YOD1 is shown to enhance the response to IL1. However, this change is also not known to recapitulate any physiological scenario.

We include new data showing that YOD1 knock-down enhances the IL-1 response in immortalized bone marrow derived macrophages (Figure 4), which provides additional evidence in a more physiological setting.

While a final definition of the physiological implication of the findings might not be reachable at this stage, the authors should be requested to provide information on two subjects that are central to any future consideration of the potential answers:

1) To explore the possibility that modulation of TRAF6 signaling by variation in the cellular level of YOD1 is physiologically relevant, the authors should compare the cellular level of YOD1 to that of TRAF6 in several different types of cells and also assess the extent of their association in these different cells.

To increase the physiological relevance, we show endogenous interaction of TRAF6 and YOD1 in different cell lines and also primary human umbilical vein endothelial cells (HUVEC) (Figure 1). Further, we compared expression and binding TRAF6 and YOD1 in HeLa, HEK293, PC3 and U2OS cells (Figure 1—figure supplement 3). We detected interaction and there was a tendency that the interaction was enhanced in cells expressing more TRAF6. Future analyses must address how YOD1 shapes TRAF6 responses, e.g. by analyzing YOD1 KO mice.

2) The need for p62 binding to TRAF6 in order to trigger signaling by the latter is not restricted to the IL-1 effect. Several other triggers that employ TRAF6, including mTOR, CD40, RANK and Nerve Growth Factor also require association of p62 with TRAF6. To confirm that the effect of YOD1 on TRAF6 activation indeed relates to its ability to interfere with p62 binding to the latter, rather than affecting some other aspect of IL-1 signaling, the authors should check if ablation of YOD1 expression facilitates signaling in response to such other triggers.

We primarily restricted the analysis to IL-1 signaling, because here the role of p62 and TRAF6 is well- documented. In line, we also show that TRAF6 and p62 knock-down strongly impairs NF-κB signaling in response to IL-1.

We have now included an analysis on the role of TRAF6, p62 and YOD1 after CD40 or RANK stimulation (Figure 6—figure supplement 1). The data yielded quite unexpected results. First, TRAF6 but not p62 knock-down significantly affected NF-κB activation in response to CD40-L (CD40 293 cells) or RANKL (PC3 cells). However, looking at the literature it becomes clear that the role of p62 in CD40 and RANK signaling is not so clear. In fact, p62 knock-down did not affect CD40 signaling in the renal adenocarcinoma cell 786-O and CD40 signaling was only slightly reduced in primary macrophages from mice carrying a p62 truncation that deletes the TRAF6 binding region (Seibold et al., BBRC 2015). Also, early RANK signaling to NF-κB was not impaired in p62 KO osteoclasts, but only a second wave of NF-κB response after several days of stimulation was abolished (Duran et al., Dev Cell 2004). Thus, the role of p62 on TRAF6 signaling is apparently highly stimulus and cell-type specific. To our surprise, in the largely p62-independent early activation of NF- κB after CD40 or RANK stimulation, YOD1 knock-down led to decreased NF-κB signaling, suggesting a positive rather than a negative effect on the signaling pathway. We include these data, because we think it is quite important to report on these findings. We included a new part in the Discussion (third paragraph). At the moment, there is no simple explanation for these results, e.g. that YOD1 is a positive regulator for TRAF6-dependent and p62-independent pathways. One interesting aspect is that YOD1 acts as a p97 co-factor. p97 was recently shown to enhance IκBα degradation and to augment NF-κB activation (Li et al., MCB 2014; Schweitzer et al., JCMM 2016). Possibly, YOD1 exerts a positive effect by facilitating p97-dependent IκBα degradation, but it is unclear why this would not be the case for IL-1 or TNF signaling. Thus, at present we think that we need to focus the study largely on the well-characterized TRAF6/p62-dependent IL-1 signaling. Future analyses must determine how YOD1 can exert opposing effects on different TRAF6 signaling pathways.

Reviewer #2:

In this study, Schimmack et al. identify the deubiquitinating enzyme YOD1 as a new interacting protein of the E3 ligase TRAF6. The authors show that YOD1 interacts with the C-terminal MATH domain of TRAF6, and inhibits TRAF6 binding with p62/Sequestosome-1. Functional studies using HeLa YOD1 overexpression and knockout cells support the idea that YOD1 is a negative regulator of IL-1-induced NF-κB activation. YOD1 appears to inhibit TRAF6 by a noncatalytic mechanism since the YOD1 C160S mutant still inhibits IL-1-NF-κB signaling. Overall, this is a very thorough and interesting study and the data are convincing for the most part. However, a weakness of the study that needs to be addressed is that much of the data were generated using overexpression strategies in cell lines (mainly HeLa and 293).

1) Almost all functional studies (stable overexpression of YOD1 and CRISPR knockouts) were performed in HeLa cells. Some of the key data should be confirmed in other cell lines, and potentially primary cells.

This is a valid point and we now included data on IL-1-induced NF-κB signaling after shRNA knock- down of TRAF6 or YOD1 in murine immortalized bone marrow derived macrophages (iBMDM) (Figure 4). In line with the data from HeLa cells, TRAF6 depletion impairs IκBα degradation, NF-κB DNA binding and NF-κB target gene expression. In contrast, downregulation of YOD1 augments NF-κB signaling, activation and target gene expression. In addition, we strengthened our findings by performing TRAF6/YOD1 co-IPs in primary human umbilical vein endothelial cells (HUVEC) (Figure 1; Figure 7). We show that YOD1 binds to TRAF6 in unstimulated cells and that binding is decreased after IL-1 stimulation. Thus, the new data provide support for a functional relevance in immune and primary cells.

2) It is surprising that mutation of the putative TRAF6 interaction motif in YOD1 does not abolish the interactions between YOD1 and TRAF6. Have the authors examined if additional mutations (e.g. P94A) in YOD1 impair TRAF6 binding?

We were also surprised by this finding and we performed a more rigorous mutagenesis. Indeed, a complete mutation of the TRAF6 binding motif on YOD1 (PPECLD to AAAGVA) did not result in a loss of TRAF6 binding and we show these data now in Figure 1—figure supplement 3.

3) There is significant potential for artifacts with overexpression of fluorescently tagged proteins in Figure 2 and 3. The authors should attempt imaging studies with endogenous YOD1, p62 and TRAF6 proteins to confirm the key results.

We agree that it would be favorable to confirm the localization by performing immunofluorescence microscopy of endogenous proteins. Unfortunately, we have not been able to detect a specific signal for YOD1 with the available antibodies in immunofluorescence stainings. We used YOD1 knock-down and knock-out cells to control the staining, but it was not possible to obtain a specific signal for YOD1. Nevertheless, we would like to point out that we detect changes in the localization depending on the co-expression (e.g. YOD1 alone versus YOD1/TRAF6 or p62/TRAF6 versus p62/TRAF6/YOD1), revealing that the speckles are not mere overexpression artefacts that are always present in the samples. Further, we confirmed localization studies by co-IPs using overexpressed and endogenous proteins as well as functional read-outs, e.g. TRAF6-catalyzed ubiquitination.

4) The EMSAs in Figure 5 and Figure 6 could be improved by the inclusion of an Oct-1 probe as a control.

We have included Oct-1 control for all EMSAs (Figure 4, Figure 5, Figure 5—figure supplement 1, Figure 6, Figure 6—figure supplement 1, Figure 6—figure supplement 1).

5) In Figure 7, the authors conclude from this data that YOD1 is released from TRAF6 after 15 min. of IL-1 stimulation. However, there is also less YOD1 expressed in the lysates at this time point. A loading control (e.g. β-actin) is needed here.

We included GAPDH as a loading control. In addition, to confirm the finding that TRAF6/YOD1 binding is diminished upon IL-1 stimulation, we repeated the experiment in primary HUVEC cells where we also show that YOD1 binding is decreased after 20 min of IL-1b stimulation (Figure 7).

6) For experiments examining the ubiquitination of endogenous TRAF6 and NEMO (e.g. Figure 7), the authors should also blot with K63 linkage specific Ub antibody. The Ub blot for Figure 7—figure supplement 1 is of poor quality and should be repeated.

Unfortunately, despite considerable efforts, we have not been able to obtain a clear signal using the anti-K63-antibody (Millipore) after stimulation and endogenous IP (anti-TRAF6 or NEMO). As shown in Figure 7—figure supplement 1, we see a rather weak K63-Ub signal under conditions of massive poly-ubiquitination after TRAF6 and p62 overexpression. Thus, in our hands the antibody is not sufficient to reliably detect K63 ubiquitin chains on endogenous signaling mediators. For this reason we performed UbiCrest to confirm by the K63-specific DUB AMSH that we obtain K63 chains in response to IL-1 stimulation (Figure 7—figure supplement 1). The Western Blot in the UbiCrest looks different and the quality is not quite as strong, because the signals are decreased after the additional DUB incubation time (30 min at 37°C). However, the experiment clearly shows that only the promiscuous DUB USP2 and the K63-specific DUB AMSH significantly decreased TRAF6- ubiquitination. Since it is well documented that TRAF6 is generating K63 ubiquitin chains and these chains cannot be efficiently hydrolyzed by YOD1, we feel that this is sufficient to support our assumption that YOD1 acts largely through a non-catalytic mechanism.

Reviewer #3:

The deubiquitinase YOD1 (OTUD2) was identified in a two-hybrid screen with TRAF6 and confirmed to interact by co-immunoprecipitation from lysates of unstimulated Hela cells. YOD1 was shown to compete for TRAF6 binding to p62/sequestosome-1, an interaction that is required for IL-1R signaling. Knockdown and overexpression of YOD1 suggested that YOD1 functions as a negative regulator of TRAF6/p62-dependent activation of NF-κB by IL-1β.

Given the importance of IL-1β in inflammation and immune responses, the regulation of IL-1R signaling is of considerable interest. The study by Schimmack et al. contains interesting new information on this subject. However, the quality of some of the results needs to be improved and quantification of some of the data plus statistical testing for significant differences is essential to support the proposed model.

As suggested, we present additional data and we performed quantification to support the proposed model (see detailed responses).

Specific points:

1) To test the specificity of YOD1 interaction with TRAF6, a panel of E3 ligases and other Ub regulatory proteins was tested for interaction with YOD1 by yeast two-hybrid assay (Figure 1—figure supplement 1). Do the authors have evidence that the constructs used for this panel actually work in a yeast two-hybrid assay?

We included a Western Blot to show expression of the constructs in yeast (Figure 1—figure supplement 1). We have generated the activation domain (AD) library to search for interactors of the Ub system in yeast. It is not feasible to control interactions for all proteins in such a panel, but in Figure 1—figure supplement 1we show some examples, e.g. expected interaction of cIAPs with UBC13 or HOIP with OTULIN. As can be seen, strong protein expression in WB is not a good predictor for functional interaction in the growth assays, because despite the low expression of TRAF6 and cIAPs the expected interactions are clearly detectable.

2) In Figure 1, HA-TRAF6 is shown to co-immunoprecipitate with FLAG-YOD1. However, the levels of HA-TRAF6 { ± } FLAG-YOD1 co-expression are very different, so it is difficult to assess whether this interaction is specific. This result should be replaced with another experiment in which HA-TRAF6 levels are more equivalent { ± } FLAG-YOD1.

As suggested we replaced the experiment by another IP.

3) In Figure 1, it is not clear whether cells have been transfected with empty vector (EV) in the third lane from the left. There appears to be a small amount of FLAG-YOD1 in the HA immunoprecipitates in this lane, although the cell lysates appear to have less FLAG-YOD1. Is the co-immunoprecipitation of FLAG-YOD1 specific?

We apologize for not mentioning that empty vector controls (HA or FLAG) were always transfected in the control lanes. The lower Western Blot was actually quenched due to too much loading and we replaced it by a new Flag-YOD1 Western Blot.

4) The level of HA-TRAF6 expressed in the left-hand lane co-transfected with EV (Figure 1) is much lower than the lysates co-expressing FLAG-YOD1 proteins. Consequently, it is difficult to assess the specificity of HA-TRAF6 co-immunoprecipitation with FLAG-YOD1.

We repeated the control Western Blot and indeed we see slightly less HA-TRAF6 in the EV control lane. However, the better control is shown in the third lane (Flag-YOD1 130-384) and here the amounts of TRAF6 are equivalent and binding is reduced, confirming the specificity of the IP.

5) siRNA knockdown of YOD1 is used to show the specificity of endogenous YOD1 co-immunoprecipitation with TRAF6 (Figure 1). In reality, this experiment simply confirms the identity of the YOD1 and demonstrates that there is less YOD1 present in the cell lysates to co-immunoprecipitate with TRAF6. This experiment should be replaced with one in which endogenous TRAF6 is knocked down and the effect on YOD1 co-immunoprecipitation with anti-TRAF6 tested.

We agree that siYOD1 is simply confirming the specificity of the signal and instead we now show TRAF6 knock-down in HeLa cells (Figure 1—figure supplement 3) and more IPs using IgG control in several cell lines (HEK293, HeLa, U2OS) and primary HUVEC (Figure 1) to confirm the endogenous TRAF6/YOD1 interaction.

6) It would strengthen the paper if endogenous TRAF6 and YOD1 could be shown to associate in cell lysates from primary human cells (e.g. HUVECs).

As suggested, we performed TRAF6/YOD1 co-IP in HUVEC to confirm interaction in primary cells (Figure 1). Further, we also show in HUVEC that TRAF6/YOD1 binding is decreased after 15-20 min of IL-1 stimulation (Figure 7).

7) The authors state that GFP-YOD1 and RFP-TRAF6 co-localize to a large extent when co-expressed (Figure 2). This conclusion would be stronger if the extent of co-localisation was quantified in multiple cells and tested for statistical significance.

For co-localization studies we used automated confocal immunofluorescence microscopy (Operetta PerkinElmer). The device and software gives unbiased multi-parameter data. We showed representative images and we plotted the intensity profiles of the different fluorophores (RFP and GFP in this case) in distinct sections, which we feel is a good method to indeed show co-localization. In fact co-localization in spots expressing RFP-TRAF6 and GFP-YOD1 was seen in all co-transfected cells. To make this more convincing, we show more pictures in Figure 2—figure supplement 1. For a quantitative analysis of co-localization we performed automated image analysis of ~200 co- transfected cells and determined the fluorescence intensities (FI) of RFP-TRAF6 and GFP-YOD1 in the GFP and RFP clusters, respectively (Figure 2—figure supplement 1). By this type of analysis we quantify the FI shown in the plot profiles in many cells. Compared to the background, the RFP-TRAF6 signal was enriched in GFP-YOD1 spots and vice versa the GFP-YOD1 signal was enhanced in RFP- TRAF6 spots, clearly suggesting co-localization of TRAF6 and YOD1 in the clusters. The data show co-localization of fluorescence signals in many cells. Unfortunately, a statistical analysis of these data is not possible. However, the additional representative pictures in combination with such an unbiased analysis convincingly show co-localization of TRAF6 and YOD1 in cellular clusters.

8) How representative are the results shown in Figure 3? Imaging results should be shown from multiple cells. If possible imaging data should be quantified and colocalization/non-colocalization tested statistically.

Co-localization of TRAF6 and p62 to cytosolic speckles is well documented (Sanz, EMBO J 2000; Seibenhener, MCB 2004; Wang, JCS 2006). We now also provide automated FI analyses of RFP- TRAF6 signal in CFP-p62 spots and vice versa to validate the co-localization by an unbiased method (Figure 3—figure supplement 1). For YOD1 and p62 co-transfection we included additional representative pictures and FI profiles to convincingly show a lack of co-clustering (Figure 3—figure supplement 1). Automated FI analysis based on defining specific YOD1 and p62 clusters was not possible in this setting, because the software was unable to define clear spots that could be analyzed for co-clustering. However, the additional representative pictures and FI profiles support our findings.

9) The authors claim that YOD1 expression decreases the interaction of TRAF6 with p62 (Figure 3). However, the level of p62 in the lysates is reduced in cells co-expressing YOD1. To convincingly demonstrate that YOD1 prevents binding of p62 to TRAF6, p62 levels in lysates need to be similar { ± } YOD1 co-expression. In addition, data need to be quantified from multiple experiments and differences tested for statistical significance.

We repeated the experiment shown in old Figure 3 with an experiment that shows less fluctuation in transfected proteins (now Figure 3). Also, we quantified the amount of p62 and YOD1 bound to TRAF6 after double transfection (control, set to 1) and triple transfection (TRAF6, p62 and YOD1) in three independent experiments (Figure 3). For quantification, the amounts of YOD1 and p62 were normalized to expression in the lysates and to co-precipitated TRAF6 to adjust for differences in transfection and IP efficiencies. The results confirm that YOD1 and YOD1 C160S displace p62 from TRAF6.

10) Based on microscopic examination of cells expressing RFP-TRAF6, Crimson-p62 and GFP-YOD1 (Figure 3), the authors conclude that p62 was excluded from YOD1/TRAF6 clusters. Only images of single cells are shown. How representative are these data (see point 8)?

To show that TRAF6 and YOD1 preferentially co-localize in triple transfected cells (RFP-TRAF6, GFP- YOD1 and Crimson-p62) we show more merged example pictures in Figure 3—figure supplement 2. As done for TRAF6/YOD1 and TRAF6/p62 automated FI analysis we compared the RFP-TRAF6 signal in RFP-TRAF6, GFP-YOD1 or p62-Crimson spots in ~ 100 triple transfected cells (Figure 3—figure supplement 2. In this unbiased analysis we found enhanced RFP-TRAF6 in GFP-YOD1 spots and to a lesser extent in Crimson-p62 spots. Vice versa, there was no enhanced signal of Crimson- p62 detected in RFP-TRAF6 or GFP-YOD1 spots over background. Taken together, the additional pictures and quantitative analysis confirms that in triple transfected cells TRAF6 preferentially co- clusters with GFP-YOD1.

11) Using HeLa cells that inducibly express YOD1 after doxycycline treatment, demonstrate that both WT and C160S YOD1 co-immunoprecipitate with TRAF6 (Figure 4). Control IgG immunoprecipitates are required to confirm the specificity of this interaction. The authors claim that IL-1β stimulation reduces YOD1-TRAF6 association. However, it appears that IL-1β stimulation actually increases association of WT YOD1 with TRAF6 while reducing that with YOD1[C160S]. The authors should comment on this difference and confirm reproducibility of these results by quantitation from multiple experiments and statistical testing.

We have removed this panel. Indeed, it seemed likely that in this overexpression scenario there was slightly augmented association of TRAF6 and YOD1 WT after short IL-1 stimulation. However, we were unable to quantify this effect. Also, despite a strong overexpression, endogenous YOD1 is still present and we have not used a tagged YOD1 version. Thus, it is not strictly clear whether there is still residual endogenous YOD1 bound or an influence of the endogenous in the case of YOD1 C160S. We show decreased binding of YOD1 to TRAF6 after IL-1 stimulation in two reconstituted HeLa KO cell clones (Figure 5 and Figure 5—figure supplement 1) and of endogenous TRAF6/YOD1 in HeLa cells and HUVEC (Figure 7), providing evidence that interaction is reduced after IL-1 stimulation.

12) Based on the results shown in Figure 4, the authors conclude that overexpression of WT or C160S YOD1 reduces IL-1β activation of three NF-κB-dependent genes. Consistent with this result, shRNA knockdown of YOD1 increases expression of the same three genes. Does doxycycline treatment alone (without YOD1 expression) alter IL-1β activation of NF-κB in the parental cells?

We have performed this experiment and show in Figure 4—figure supplement 1that DOX does not significantly affect target gene expression in parental cells.

13) Figure 6 shows the inhibitory effect of TRAF6 knockdown on IL-1β induced expression of Nfkbia and Tnfaip3 mRNAs. Does p62 knockdown affect the expression of these genes?

We did not include qRT-PCR from p62 knock-down cells, because RPII RNA used for normalization was affected by sip62. We now repeated the experiment using 18S rRNA for normalization to show that also p62 impairs induction of NF-κB target genes and the results are shown in Figure 6—figure supplement 1.